# Sustainable production of hydrogen with high purity from methanol and water at low temperatures

Sai Zhang [1,2], Yuxuan Liu[3], Mingkai Zhang[3], Yuanyuan Ma[1], Jun Hu [4] ✉ & Yongquan Qu [1] ✉

Carbon neutrality initiative has stimulated the development of the sustainable methodologies for hydrogen generation and safe storage. Aqueous-phase reforming methanol and $H_2O$ (APRM) has attracted the particular interests for their high gravimetric density and easy availability. Thus, to efficiently release hydrogen and significantly suppress CO generation at low temperatures without any additives is the sustainable pursuit of APRM. Herein, we demonstrate that the dual-active sites of Pt single-atoms and frustrated Lewis pairs (FLPs) on porous nanorods of $CeO_2$ enable the efficient additive-free $H_2$ generation with a low CO (0.027%) through APRM at 120 °C. Mechanism investigations illustrate that the Pt single-atoms and Lewis acidic sites cooperatively promote the activation of methanol. With the help of a spontaneous water dissociation on FLPs, Pt single-atoms exhibit a significantly improved reforming of *CO to promote $H_2$ production and suppress CO generation. This finding provides a promising path towards the flexible hydrogen utilizations.

Developing hydrogen ($H_2$) generation and storage technologies to sustainably supply such a clean power energy resource with high calorific value is crucial to alleviating the global energy/environmental crisis and realizing the carbon neutrality initiative[1,2]. However, the inflammability and explosibility of $H_2$, as well as its extremely high liquefied pressure (700 psi), technically and economically raise the grand challenges in cost, safety, and reliability during the operations for hydrogen transportation and storage. Afterward, various liquid organic carriers have been proposed to store hydrogen in the dense liquid phase, which offers a promising methodology with high capacity and safe re-distributions on demands. Among various liquid hydrogen carriers (formic acid[3–5], *N*-heteroarenes[6–8], cyclohexane[9], etc.), methanol, as a sustainable, inexpensive, and readily available hydrogen source derived from biomass and/or $CO_2$ hydrogenation, can give 18.8 wt.% $H_2$ gravimetric density via reforming with $H_2O$ to release three equivalent amounts of $H_2$. Combining its mature and safe technology for storage and transportation, methanol has been recognized as one of the most promising candidates among all known liquid hydrogen carriers[10,11].

Although a homogeneous metal catalytic system with meticulously designed ligands can enable hydrogen release at temperatures below 100 °C, the large amounts of strong bases (e.g., 8 M KOH, NaOH) are generally adopted to trigger the entire catalytic process by activating $CH_3OH/H_2O$ at low temperatures and maintain the catalytic activity by neutralizing the generated formic acid as well as $CO_2$[10,12–14]. From the environmentally sustainable and economical perspectives, various heterogeneous catalysts, including Pt-, Ru-, Pd-based catalysts, have been developed as the additive-free catalytic system for hydrogen generation. However, the reforming of methanol and water by those catalysts generally faces two big obstacles: (1) the high temperatures (å 250 °C) to boost catalytic reaction, and (2) the low purity of $H_2$ accompanied by the generation of CO at a high level[15–19]. Recent advances in developing new heterogeneous catalysts have greatly decreased the operation temperatures as low as 150 °C for the

[1]School of Chemistry and Chemical Engineering, Northwestern Polytechnical University, 710072 Xian, China. [2]Research & Development Institute of Northwestern Polytechnical University in Shenzhen, 518057 Shenzhen, China. [3]Center for Applied Chemical Research, Frontier Institute of Science and Technology, Xian Jiaotong University, 710049 Xian, China. [4]School of Chemical Engineering, Northwest University, 710069 Xian, China. ✉e-mail: hujun@nwu.edu.cn; yongquan@nwpu.edu.cn

aqueous-phase reforming of methanol by using the atomically dispersed Pt on α-MoC[11]. Afterward, the further decrease of the reaction temperatures with a satisfactory $H_2$ generation rate is extremely difficult and rarely realized on heterogeneous catalysts yet up to now. Therefore, developing high-efficient catalysts capable of in situ releasing of $H_2$ at even lower temperatures and the suppressed CO generation is highly desirable for the large-scale production of hydrogen, bringing us a step closer to a methanol economy.

Herein, we demonstrate that the dual-active site catalysts composed of the Pt single-atoms and frustrated Lewis pairs (FLPs) on the atomically dispersed Pt anchored on porous nanorods of $CeO_2$ ($Pt_1$/PN-$CeO_2$) enable a stabilized $H_2$ generation at a low temperature of 120 °C through a base-free aqueous-phase reforming of methanol (APRM). The catalytic activity and selectivity were examined in a closed system under an initial pressure of 0.4 MPa $N_2$. The turnover frequency (TOF) of $Pt_1$/PN-$CeO_2$ based on Pt atoms was 33 $h^{-1}$ at 120 °C, which was comparable to or even superior to the majority of homogeneous and heterogeneous catalysts operated in the presence/absence of additives/bases under similar reaction temperatures. Encouragingly, the selectivity of CO by-product, a harmful molecule for downstream metal catalysts, was significantly suppressed below 0.03% under such mild reaction conditions.

## Results

### Catalyst design

To achieve high activity of catalysts for reforming methanol and water at low temperatures, the key steps are analyzed initially. Theoretically, $H_2$ generation from methanol and water generally involves three reaction steps: (1) $CH_3OH$ dissociation into *CO and *H intermediates; (2) $H_2O$ activation into *OH and *H intermediates; and (3) transformation of *CO and *OH into $CO_2$(g) and *H through a water-gas shift reaction[20]. Therefore, the ideal scenario requires a catalyst with the effective activation of both $CH_3OH$ and $H_2O$ to guarantee efficient $H_2$ generation at low temperatures. However, there is difficult to be achieved on a single type of active site[11,19]. The principle behind this phenomenon is that the metal active sites with high capability for $CH_3OH$ activation generally exhibit a low ability for $H_2O$ dissociation[20,21]. Different from $CH_3OH$ activation on metal surfaces, the $H_2O$ activation can be significantly promoted on reducible metal oxides via O atom interaction with Lewis acidic center (metal site), and then one of H atoms transfers to the adjacent Lewis basic center (lattice oxygen site, Fig. 1a)[20,22–26]. Therefore, the construction of the dual-active sites for the respective methanol and water activation potentially provides a feasible approach to efficiently producing $H_2$ at low temperatures.

Of the candidate catalysts for $CH_3OH$ activation, Pt is selected due to its strong capacity for the $CH_3OH$ dissociation into *CO and *H active intermediates[11,27]. Then, density function theory (DFT) calculations were performed to investigate the $H_2O$ activation on Pt (111) surface. As expected, the Pt(111) surface exhibited the kinetically unfavorable process for $H_2O$ activation with a derived large kinetic energy barrier of 1.12 eV (Fig. 1b, c)[11]. Therefore, similar to other metals[22], the $H_2O$ dissociation is difficult to occur on Pt surface, thereby limiting the subsequent reforming of the *CO intermediates with *OH at low temperatures.

Inspired by the typical Lewis acid-base interaction for $H_2O$ activation (Fig. 1a), frustrated Lewis Pairs (FLPs), which are composed of a sterically hindered Lewis acid and Lewis base[28–31], provides a promising opportunity for efficiently activating $H_2O$ molecule at low temperatures. In this configuration, the negative O and H atoms of $H_2O$ interact closely with Lewis acidic site and basic site of FLPs at the same time (Fig. 1a), respectively, revealing an attractive pathway for $H_2O$ activation. Recently, the all-solid FLP sites constituted by the two adjacent surface $Ce^{3+}$ (Lewis acid sites) and the neighboring surface lattice oxygen (Lewis base sites) have been successfully constructed on the

PN-$CeO_2$, as shown in Supplementary Fig. 1. Due to the unique configuration of FLPs and the different formation energies of oxygen vacancy on various surfaces, $CeO_2$(110) surface instead of $CeO_2$(100) and $CeO_2$(111) surfaces has been previously verified to exhibit the highest probability for the FLP's construction (Supplementary Fig. 2)[32,33]. Those investigations have demonstrated that the abundant surface defects of oxygen vacancy on $CeO_2$(110) benefit the formation of interfacial FLP sites. To explore their ability for $H_2O$ activation, DFT calculations were thereafter performed to investigate the $H_2O$ decomposition on $CeO_2$(110) surfaces with various densities of a structural defect, i.e., ideal $CeO_2$(110), $CeO_2$(110) with one oxygen vacancy ($CeO_2$(110)-$V_O$) and $CeO_2$(110) surface with FLPs sites ($CeO_2$(110)-FLP).

On the ideal $CeO_2$(110) surface, the $H_2O$ activation can easily occur with a low energy barrier of 0.58 eV (Fig. 1d, e). More importantly, the dissociation of $H_2O$ can be further promoted in the presence of oxygen vacancy on $CeO_2$(110) surface. As shown in Fig. 1b, c, both $CeO_2$(110)-$V_O$ with one oxygen vacancy and $CeO_2$(110)-FLP with two adjacent oxygen vacancies exhibit the adsorptive dissociation of $H_2O$ without a transition state in comparison with ideal $CeO_2$(110). Especially, the dissociation of $H_2O$ on $CeO_2$(110)-FLP is also the thermodynamically favored process compared with $CeO_2$(110)-$O_V$ active sites. Therefore, both kinetic and thermodynamic analysis reveals the FLP sites with the optimal ability to dissociate $H_2O$.

Considering that the single-atom Pt active sites exhibit higher capability for $CH_3OH$ activation than Pt nanoparticles[11,27], the construction of the dual-active sites of single-atom Pt and FLPs ($Pt_1$-FLP) can effectively activate both $H_2O$ and $CH_3OH$, respectively, potentially reducing the reaction temperatures of APRM. Then, DFT calculations were further used to explore the possible spatial structures of Pt atom on $CeO_2$(110) surface, which delivered two configurations. As shown in Supplementary Fig. 3, the Pt atom prefers to occupy the oxygen defect of $CeO_2$(110) surface owing to the lowest formation energy (1.35 eV). In this configuration, the FLP site is not affected by the single-atom Pt in the distance. Therefore, the Type **I** of $Pt_1$-FLP due-active site is successfully constructed, in which Pt single-atom locates at the oxygen vacancy of $CeO_2$(110) surface (Fig. 1d). In addition, the Pt single-atom can occupy one of the oxygen vacancies adjacent to the FLP site with slightly high formation energy (1.70 eV). For this configuration, the type **II** of $Pt_1$-FLP dual-active site is spatially adjacent to each other, as shown in Fig. 1d and Supplementary Fig. 3. Also, the spatial and electronic configuration of FLP sites are preserved in the presence of the nearby single Pt atom.

Due to the high ability for $H_2O$ activation on FLP sites, the reforming of *CO with *OH on the $Pt_1$-FLP dual-active site is also significantly improved through a water-gas shift reaction process. As shown in Supplementary Fig. 4, the energy barrier is only 0.89 eV on the adjacent $Pt_1$-FLP dual-active site (type **II**), which is dramatically lower than that of 2.15 eV on the surface of Pt(111) surface. For the type **I**, due to the long spatial distance between Pt single-atoms and FLPs sites, the generated *OH on FLP sites could diffuse near the Pt single-atom owing to the easily occurred migration of H and O atoms on $CeO_2$(110)[34–36]. Theoretically, the dual-active sites of single-atom Pt and FLPs (Fig. 1d) can significantly reduce the reaction temperature for APRM, resulting in successful $H_2$ generation at low temperatures.

### Synthesis and characterizations of $Pt_1$-FLP dual-active site catalysts

The open question herein is how to synthesize the $Pt_1$-FLP dual-active sites to devise an efficient catalyst for $H_2$ generation from APRM. A two-step hydrothermal process was used to prepare highly defective PN-$CeO_2$ with a length of ~65 nm and a daimeter of ~7 nm, as revealed from the dark-field transmission electron microscopy (TEM) image (Supplementary Fig. 5a). The 0.275 nm of lattice frings spacing was consistent with the (220) crystal face of $CeO_2$ (Supplementary Fig. 5b),

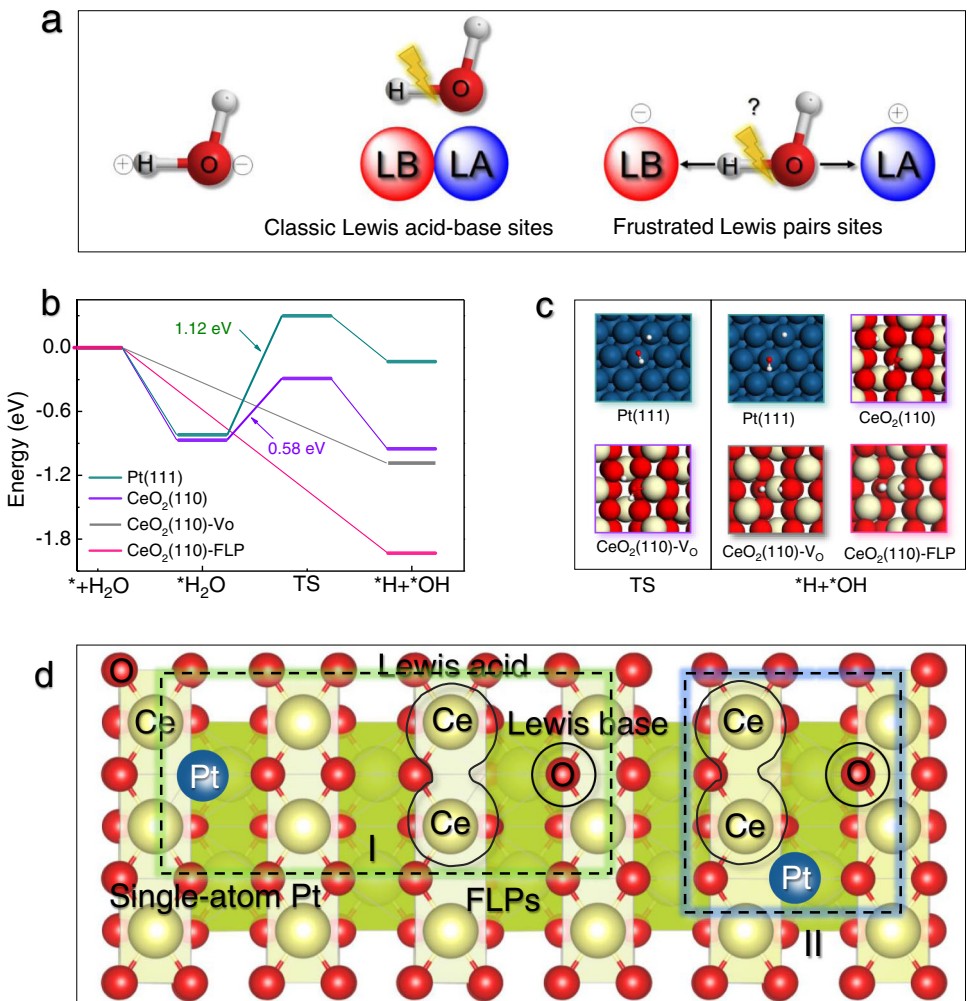

**Fig. 1 | Catalyst design. a** Scheme of $H_2O$ activation on classic Lewis acid-base sites and FLPs. **b** Energy barrier and **c** adsorption configuration of $H_2O$ activation on various actives sites. **d** The designed dual-active sites of single-atom Pt and FLPs.

Note: The dark bule, red, and yellow balls respective the Pt, O, and Ce atoms, respectively. The abbreviation of FLP is the frustrated Lewis pairs.

further revealing PN-CeO$_2$ along with [110] direction[37]. The specific surface area of PN-CeO$_2$ was 109 m$^2$ g$^{-1}$, as derived from the N$_2$ adsorption/desorption isotherm plot (Supplementary Fig. 6a). The porous structure with a size of 1.5–3.0 nm was revealed from TEM images (Supplementary Fig. 5b) as well as the Brunauer-Emmett-Teller (BET) measurements (Supplementary Fig. 6b, c). More importantly, the abundance of surface defect on PN-CeO$_2$ was indexed by the 30.8% surface Ce$^{3+}$ fraction as well as the 47.1% surface Ce$^{3+}$-O fraction, derived from its X-ray photoelectron spectroscopy (XPS) spectrum of Ce *3d* and O *1s*, respectively (Supplementary Fig. 7 and Supplementary Table 1). Therefore, the FLP sites could be formed on the PN-CeO$_2$ supports owing to the high concentration of oxygen defect on the CeO$_2$(110) surface, as described in our previous reports[32,33].

Then, the single-atom Pt anchored on PN-CeO$_2$ (Pt$_1$/PN-CeO$_2$) with 0.36 wt.% loading was successfully synthesized through a photo-assisted deposition process due to the strong trapping of metal species on the defective sites of PN-CeO$_2$ from the DFT calculations (Supplementary Fig. 3). Both the specific surface area/pore structure (Supplementary Fig. 6) and levels of surface oxygen defects (Supplementary Fig. 7) of PN-CeO$_2$ were preserved during the photo-assisted Pt deposition process, indicating the maintained surface FLP sites. According to the aberration-corrected high-angle annular dark-field scanning transmission electron microscopy (HAADF-STEM) image (Fig. 2a), the single-atom Pt in catalysts was experimentally

demonstrated, which was further verified from the uniform Pt distribution on PN-CeO$_2$ by the energy dispersive spectroscopy (EDS) mapping (Fig. 2b). X-ray absorption near edge structures (XANES) of Pt K-edge revealed that the white line peak of the Pt$_1$/PN-CeO$_2$ catalysts located at 11,568.7 eV (Fig. 2c), which was very close to that of PtO$_2$. The $k^3$-weight Fourier transforms of extended X-ray absorption fine structure (EXAFS) spectra of Pt$_1$/PN-CeO$_2$ delivered one prominent peak at ~1.63 Å, which was labeled as Pt–O bond (Fig. 2d). Also, the lack of Pt–Pt coordination again suggested no Pt particles and clusters in Pt$_1$/PN-CeO$_2$ (Fig. 2d and Table 1), indicating the atomically dispersed Pt supported on PN-CeO$_2$.

Then, the dispersion and chemical environments of Pt on PN-CeO$_2$ were studied by diffuse-reflectance infrared Fourier-transform spectroscopy (DRIFTS). The peak at ~2090 cm$^{-1}$ was assigned to the linearly adsorbed CO on isolated ionic Pt$^{2+}$ (Supplementary Fig. 8)[38–40], revealing that the Pt active sites existed as atomic dispersion on the surface of PN-CeO$_2$ and coordinated with O atoms. Specifically, the two apparent peaks at 2099 and 2076 cm$^{-1}$ could be attributed to the single-atom Pt located in the configurations of the type **I** and type **II** (Fig. 1d), respectively, owing to the lower valence state of Pt on oxygen vacancy adjacent to the FLP sites than it on oxygen vacancy in the distance (Supplementary Fig. 3). Therefore, combining with the HAADF-STEM, XANES, and DRIFTS results, the dual-active sites of single-atom Pt and FLPs were successfully constructed on the surface

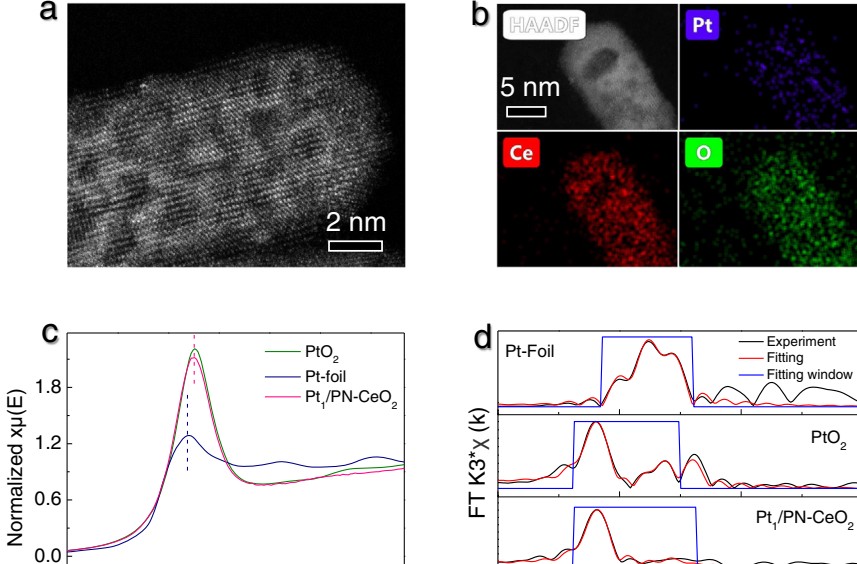

**Fig. 2 | Characterizations of the Pt₁/PN-CeO₂ catalyst. a** HAADF-STEM image and **b** EDS mapping. **c** XANES spectra of Pt-foil, PtO₂, and Pt₁/PN-CeO₂. **d** The $k^3$-weighted Fourier-transformed spectra derived from the EXAFS spectra of the Pt₁/PN-CeO₂ catalysts and PtO₂.

of Pt₁/PN-CeO₂ catalysts, where the single-atom Pt mainly occupied oxygen vacancy.

## Catalytic performance for H₂ generation

Then, the catalytic H₂ generation of Pt₁/PN-CeO₂ through APRM methodology was evaluated under base-free conditions at various temperatures. To highlight the advantages of Pt₁/PN-CeO₂, Pt nanoparticles anchored on Al₂O₃, TiO₂, and C with a loading of 0.5 wt.% were also prepared as the reference catalysts by impregnation method (Supplementary Fig. 9). At the reaction temperature of 135 °C, the Pt₁/PN-CeO₂ catalysts yielded a H₂ generation rate of 199 $mol_{H_2}$ $mol_{Pt}^{-1}$ $h^{-1}$, which was two or three orders of magnitude enhancement in the H₂ generation rates catalyzed by Pt/Al₂O₃ (2.6 $mol_{H_2}$ $mol_{Pt}^{-1}$ $h^{-1}$), Pt/TiO₂ (3.8 $mol_{H_2}$ $mol_{Pt}^{-1}$ $h^{-1}$), and Pt/C catalysts (0.7 $mol_{H_2}$ $mol_{Pt}^{-1}$ $h^{-1}$) under the identical conditions, respectively (Fig. 3a). Even when the reaction temperature was reduced to 120 °C, the Pt₁/PN-CeO₂ catalysts still delivered a high H₂ generation rate of 33 $mol_{H_2}$ $mol_{Pt}^{-1}$ $h^{-1}$ (Fig. 3a). Comparatively, bare H₂ was detected at this temperature for Pt catalysts anchored on other functional supports herein as well as in the previous reports[15,16,41,42]. In addition, the H₂ generation of 19.7 $mol_{H_2}$ $mol_{Pt}^{-1}$ $h^{-1}$ could be obtained by the fixed bed at a temperature as low as 100 °C. Most importantly, such a H₂ generation rate catalyzed by Pt₁/PN-CeO₂ is comparative to and even better than the H₂ generation

rates when noble metal homogeneous catalysts are used in the presence of a high concentration strong base as additives at similar reaction temperatures (Fig. 3b and Supplementary Table 2)[12,13,43–45]. To the best of our knowledge, it is among one of the lowest temperatures for all previously reported heterogeneous metal catalysts to achieve the efficient H₂ generation of reforming of methanol and water in the absence of any additives.

Most impressively, the selectivity of CO by-product catalyzed by Pt₁/PN-CeO₂ was only 0.032% and 0.027% at the reaction temperatures of 135 and 120 °C, respectively. When the reaction temperature was further increased to 165 °C, the H₂ generation rate was significantly enhanced to 1103 $mol_{H_2}$ $mol_{Pt}^{-1}$ $h^{-1}$. Encouragingly, the selectivity of CO was still maintained as low as 0.045% at this high reaction rate. Therefore, the Pt₁/PN-CeO₂ catalysts exhibited excellent catalytic performance for H₂ generation from methanol and water, especially at low reaction temperatures.

In addition, the catalytic stability of Pt₁/PN-CeO₂, a critical factor of heterogeneous catalysts, was also evaluated through a cycling test at 165 °C (1 h per cycle) and 120 °C (3 h per cycle), respectively. At 165 °C, the H₂ generation was slightly decreased from 1103 $mol_{H_2}$ $mol_{Pt}^{-1}$ to 881 $mol_{H_2}$ $mol_{Pt}^{-1}$ from the first cycle to the tenth cycle with a low CO selectivity <0.05% for each cycle (Fig. 3c). Also, the H₂ generation was well maintained at the range of 93–128 $mol_{H_2}$ $mol_{Pt}^{-1}$ with a low CO selectivity <0.03% during 10 cycles at 120 °C (Fig. 3c). The preserved morphological features of the spent Pt₁/PN-CeO₂ catalysts could further reveal the satisfactory structure stability operated at both 165 °C and 120 °C (Supplementary Fig. 10a, b). Meanwhile, the surface Ce³⁺ fraction along with the Ce³⁺-O fraction of the used Pt₁/PN-CeO₂ catalysts, was similar to those of as-synthesized Pt₁/PN-CeO₂ (Supplementary Fig. 11), revealing that the FLP sites were stable throughout the catalytic hydrogen generation. After careful analysis of the HAADF-STEM images (Supplementary Fig. 10c), the Pt nanoclusters with a small amount were observed on the surface of the used Pt₁/PN-CeO₂ catalysts at 165 °C. A previous report has proved that the perimeter Pt active sites in the Pt-CeO₂ catalytic system remain dynamically mobile for the reforming of *CO[46]. Thus, the decrease in H₂ generation rate of Pt₁/PN-CeO₂ could be attributed to the slightly increased size of Pt active sites owing to the possible mobility of the

## Table 1 | FT-EXAFS spectra and the fitting curves (without phase correction)

| Sample | Path | N | Sigma² (×10⁻³) | R (Å) | R-factor |
|---|---|---|---|---|---|
| Pt-foil | Pt–Pt | 12 (set)ᵃ | 4.71 ± 0.33 | 2.76 ± 0.01 | 0.002 |
| PtO₂ | Pt–O | 6.1 ± 1.2 | 3.3 ± 4.9 | 2.06 ± 0.07 | 0.008 |
| | Pt–Pt | 9.2 ± 2.8 | 5.6 ± 6.5 | 3.10 ± 0.84 | |
| | Pt–O 2nd shell | 11.9 ± 5.3 | 4.2 ± 7.2 | 3.48 ± 0.13 | |
| Pt₁/PN-CeO₂ | Pt–O | 5.8 ± 0.6 | 3.0 ± 1.4 | 2.01 ± 0.60 | 0.015 |
| | Pt–Pt | 0 | 0 | 0 | |

ᵃamp = 0.80 ± 0.04.

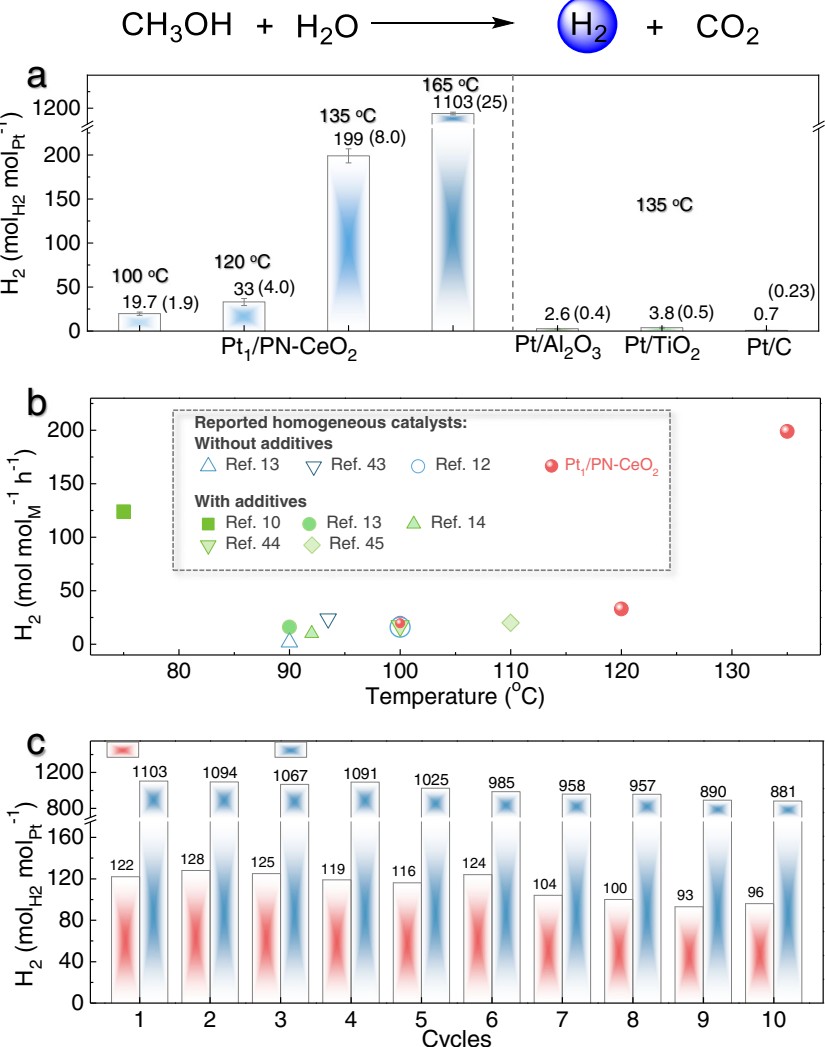

**Fig. 3 | Catalytic performance. a** $H_2$ generation from methanol and $H_2O$ catalyzed by various Pt catalysts. The error is derived from three parallel experiments. Reaction conditions: catalysts (50 mg), $CH_3OH$ (40 mL), $H_2O$ (18 mL), n($CH_3OH$):n($H_2O$) = 1:1, and $N_2$ (0.4 MPa). **b** Summary of the $H_2$ generation catalyzed by $Pt_1$/PN-CeO$_2$ in the absence of additives compared with various reported homogeneous catalysts with or without additives at low temperature. **c** Cycling of $Pt_1$/PN-CeO$_2$ for $H_2$ generation from methanol and $H_2O$. The reaction time of each cycle was 1 and 3 h at 160 and 120 °C, respectively. Reaction conditions: $Pt_1$/PN-CeO$_2$ (50 mg), $CH_3OH$ (40 mL), $H_2O$ (18 mL), n($CH_3OH$):n($H_2O$) = 1:1, and $N_2$ (0.4 MPa).

atomically dispersed Pt on the surface of catalysts. Nevertheless, combining the base-free and absence of other additives, the high $H_2$ generation rate and low CO selectivity, as well as the satisfactory long-term stability, make $Pt_1$/PN-CeO$_2$ featured as easy operation and high sustainability, exhibiting its great promises for practical $H_2$ generation from APRM at low temperatures.

## Mechanism analysis

To verify our design concept of the $Pt_1$-FLP dual-active sites for the enhanced activity and suppressed CO generation, we performed a series of control experiments to explore the catalytic mechanism of $Pt_1$/PN-CeO$_2$ for $H_2$ generation *via* APRM process. As predicted from theoretical calculations, the single-atom Pt and FLPs on CeO$_2$ can significantly activate $CH_3OH$ and $H_2O$, respectively. Thereby, the contributions of Pt and FLPs for catalytic reaction were experimentally examined. In the absence of Pt, no $H_2$ was generated from the reaction system with PN-CeO$_2$ at 120 °C or 165 °C (Supplementary Table 2, Entry 8–9), demonstrating the inert nature of FLPs as well as the critical roles of Pt for APRM herein.

Then, to further inquiry the contributions of the FLP sites on PN-CeO$_2$, the nonporous nanorods of CeO$_2$ (NR-CeO$_2$) and nanoparticles

of CeO$_2$ (NP-CeO$_2$) were also selected as the referenced supports for $H_2$ generation. We have previously demonstrated that the highly defective PN-CeO$_2$ indexed by a large surface Ce$^{3+}$ fraction exhibits a high probability to construct the interfacial FLP sites[32,47]. Thus, NR-CeO$_2$ with a lower surface Ce$^{3+}$ fraction (19.7%, Supplementary Fig. 12) exhibited less amount of surface FLP sites, in comparison with PN-CeO$_2$ (30.8%, Supplementary Fig. 7). Due to the mismatched spatial configurations and large formation energy of oxygen vacancy, NP-CeO$_2$ with the mainly exposed CeO$_2$(111) surface as well as the lowest surface Ce$^{3+}$ fraction (15.8%, Supplementary Fig. 13) exhibited the unsuccessful formation of interfacial FLP sites, as illustrated in our previous report[32].

The Pt nanoparticles supported on PN-CeO$_2$ (Pt/PN-CeO$_2$), NR-CeO$_2$ (Pt/NR-CeO$_2$), and NP-CeO$_2$ (Pt/NP-CeO$_2$) were prepared by impregnation method with a Pt-loading of 0.51 wt.%, 0.52 wt.%, and 0.51 wt.%, respectively. The similar Pt sizes of Pt/PN-CeO$_2$ (1.39 ± 0.31 nm), Pt/NR-CeO$_2$ (1.36 ± 0.41 nm), and Pt/NP-CeO$_2$ (1.31 ± 0.1 nm) excluded the size effects on $H_2$ generation (Supplementary Fig. 14). Meanwhile, the surface Ce$^{3+}$ fractions of Pt/PN-CeO$_2$ (Supplementary Fig. 7), Pt/NR-CeO$_2$ (Supplementary Fig. 12) and Pt/NP-CeO$_2$ (Supplementary Fig. 13) were similar to their corresponding

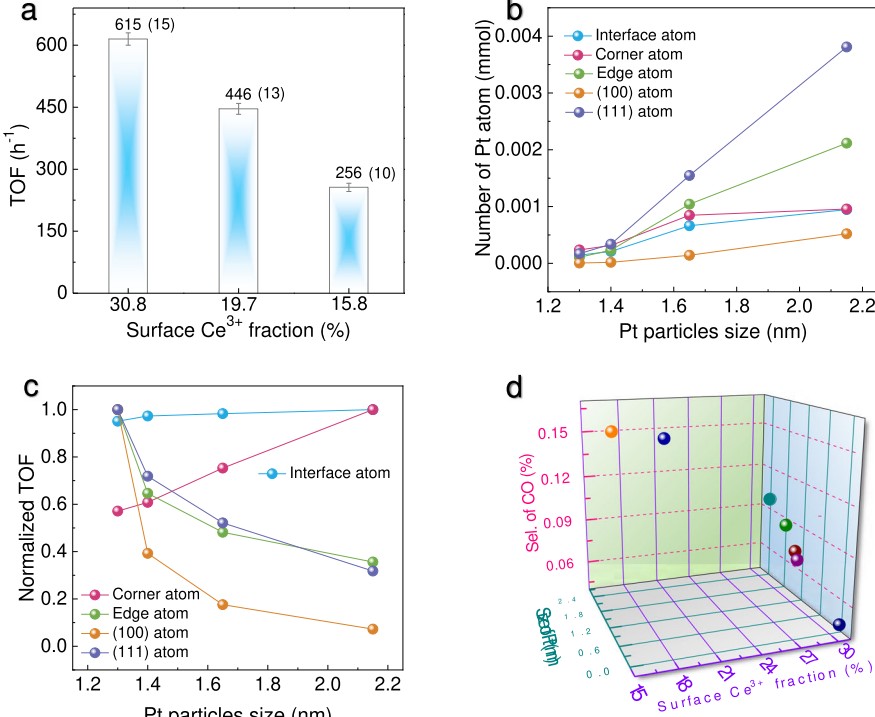

**Fig. 4 | The influence of dual-active sites on the H₂ generation performance.** **a** Plots of TOF with surface $Ce^{3+}$ fraction. The error is derived from three parallel experiments. **b** Plots of number of surface atoms per mole of Pt with Pt particle size of truncated cuboctahedron. **c** Plots of the normalized TOF values with Pt particle size. **d** The influence of Pt size and surface $Ce^{3+}$ fraction on the selectivity of CO.

supports, indicating that the amount of FLP sites on various catalysts were preserved during the impregnation process. Therefore, the Pt/PN-CeO₂, Pt/NR-CeO₂, and Pt/NP-CeO₂ catalysts exhibited the structural features of a similar Pt size, but various densities of FLPs in the order of Pt/PN-CeO₂ > Pt/NR-CeO₂ ≫ Pt/NP-CeO₂.

Among three catalysts, Pt/PN-CeO₂, with the most abundant surface FLP sites, exhibited the highest apparent catalytic activity for H₂ generation under the same reaction conditions (Supplementary Table 2, Entries 10–12), where the reaction temperature was at 165 °C. To further identify the intrinsic activity, the turnover frequency (TOF) values based on each exposed Pt atom were calculated to eliminate the influence of Pt sizes on catalytic activity. As shown in Fig. 4a, the TOF values increased significantly with the increase of the surface $Ce^{3+}$ fractions (the indicator of the surface FLPs density). Specifically, the Pt/PN-CeO₂ catalysts yielded a TOF of 615 h⁻¹, which was 1.7 and 2.4 times higher than that of Pt/NR-CeO₂ (364 h⁻¹) and Pt/NP-CeO₂ (256 h⁻¹), respectively. Therefore, the FLPs sites can greatly accelerate the APRM process.

Based on the above-controlled experiments, the excellent catalytic performance of Pt₁/PN-CeO₂ for H₂ generation *via* APRM could be attributed to cooperation between single-atom Pt and FLPs. As shown in theoretical calculations (Supplementary Fig. 4), the generated *CO on single-atom Pt undergoes a reforming with the formed *OH active species via water dissociation on FLP sites. Thus, only the interfacial Pt atom in nanoparticles may play the dominated contribution to achieve the synergistically cooperative catalytic effect for APRM. To examine this assumption, a series of Pt/PN-CeO₂ with various Pt nanoparticles sizes were prepared by the impregnation method to obtain a various number of interfacial Pt atoms on PN-CeO₂ surface. As shown in Supplementary Figs. 15 and 16, the sizes of Pt nanoparticles were increased from 1.30 ± 0.29 nm, to 1.39 ± 0.31 nm, and then to 1.65 ± 0.44 nm and finally to 2.15 ± 0.75 nm, when the loadings of Pt were controlled at 0.32, 0.51, 2.1, and 5.1 wt.%, respectively. Generally, the small Pt nanoparticles (<5.0 nm) are prone to display a morphology of truncated octahedron with the exposed (100) and (111) facets[48,49]. Then, the

number of interface, corner, edge, (111), or (100) Pt atoms over the differently sized Pt anchored on PN-CeO₂ were calculated and shown in Fig. 4b (the detailed analysis methodology in the Supplementary Information, including Supplementary Tables 4–7, and Supplementary Fig. 17).

Experimentally, the initial H₂ generation rates for those catalysts were tested at 165 °C and summarized in Supplementary Table 2 (Entries 10 and 13–15). Assuming the uniform activity of each type of Pt site regardless of their sizes, the dominated active Pt species should show a near linearly increased activity as well as a constant catalytic activity (indexed by TOF) for the H₂ generation. Then, the TOF value of each type of Pt active site was calculated (Supplementary Fig. 18) and normalized to the maximum one of each type of Pt active site. As shown in Fig. 4c, the normalized TOF values of the corner Pt increased continuously with the increased sizes of Pt particles. While, the normalized TOF values of the edge, (110), and (111) Pt decreased monotonically with the increased Pt sizes. Thus, none of them were considered the real active sites for APMR. Surprisingly, the normalized TOF values of the interface Pt were almost constant regardless of their sizes (Fig. 4c), suggesting that the interface Pt atoms were the dominated active sites for the H₂ generation through the reforming of methanol and water.

Beyond activity, the release of CO during H₂ generation was greatly suppressed in the presence of abundant FLPs sites. The FLP sites on PN-CeO₂ effectively promote the H₂O activation for the generation of *OH species (Fig. 1), which can further improve the reforming of *CO to CO₂ and thereby reduce the direct release of CO. Consistently, the CO selectivity reduced experimentally with the increased surface $Ce^{3+}$ fractions (an indicator of the number of FLPs, Fig. 4d). Importantly, the CO generation was also largely suppressed when the sizes of Pt were reduced (Fig. 4d). The generated *OH intermediates on CeO₂ are easily accessible to directly attack the *CO adsorbed on the interfacial Pt atoms, also resulting in the promoted reforming of *CO to CO₂. Therefore, the co-existence of the interfacial

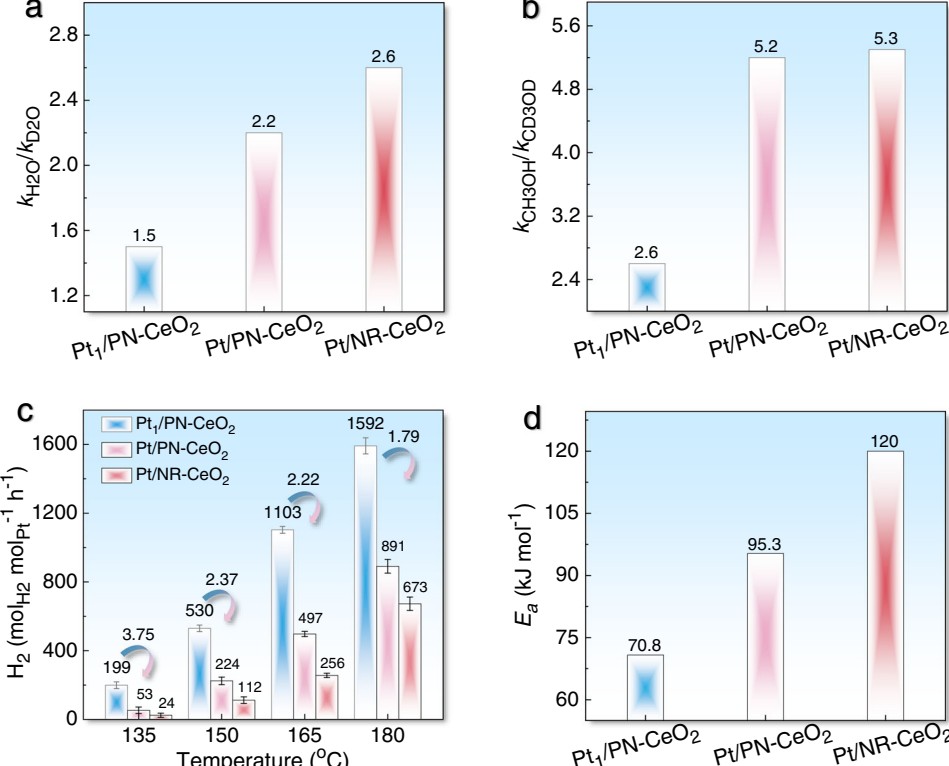

**Fig. 5 | Kinetic analysis of dual-active sites for $H_2$ generation from methanol and $H_2O$. a** Ratios of $H_2$ generation rate from $CH_3OH/H_2O$ to $CH_3OH/D_2O$. **b** Ratios of $H_2$ generation rate from $CH_3OH/H_2O$ to $CD_3OD/H_2O$. **c** $H_2$ generation rates at various reaction temperatures catalyzed by $Pt_1/PN\text{-}CeO_2$, $Pt/PN\text{-}CeO_2$, and $Pt/NR\text{-}CeO_2$. The error is derived from three parallel experiments. **d** $E_a$ of various Pt catalysts for $H_2$ generation from methanol and $H_2O$.

Pt and FLPs effectively improves the reforming of *CO to suppress the CO generation.

From the above analysis, the cooperation of interfacial Pt atoms and FLP sites is experimentally and theoretically demonstrated for the selective $H_2$ generation from methanol and $H_2O$ at low temperatures. The $Pt_1/PN\text{-}CeO_2$ catalysts with single-atom Pt and abundant surface oxygen defects fully expose the Pt atom and construct the richest interfacial $Pt_1$-FLP dual-active sites, thereafter achieving the significant promotion of $H_2$ production with satisfactory CO suppression for APRM at low temperatures.

In order to further clarify the essence of dual-active sites for each step to boost the catalytic performance of $Pt_1/PN\text{-}CeO_2$, the H/D isotope experiments were performed on the $Pt_1/PN\text{-}CeO_2$, $Pt/PN\text{-}CeO_2$, and $Pt/NR\text{-}CeO_2$ catalysts under the given reaction conditions. The $Pt_1/PN\text{-}CeO_2$ and $Pt/PN\text{-}CeO_2$ catalysts exhibited the close density of FLP sites but different amounts of interface Pt atom. While $Pt/PN\text{-}CeO_2$ and $Pt/NR\text{-}CeO_2$ with similar Pt sizes showed different densities of the interfacial FLP sites. Derived from their $H_2$ generation rates (Supplementary Fig. 19), the value of $k_{H2O}/k_{D2O}$ for $Pt/PN\text{-}CeO_2$ (2.2) was slightly lower than that for $Pt/NR\text{-}CeO_2$ (2.6). Due to the small fraction of interfacial Pt atoms, the promotion of FLPs sites on $PN\text{-}CeO_2$ could be vaguely embodied by the nanoscaled catalysts. When the metal sites are all interfacial Pt atoms as featured in $Pt_1/PN\text{-}CeO_2$, the promotion by FLPs for $H_2O$ activation was significantly enhanced, as revealed from the lowest value of $k_{H2O}/k_{D2O}$ (1.5). This phenomenon further indicated that the interfacial Pt atoms were the dominated active sites for the $H_2$ generation *via* the accelerated reforming of *CO and *OH.

Besides, the $Pt/PN\text{-}CeO_2$ and $Pt/NR\text{-}CeO_2$ catalysts exhibited similar values of $k_{CH3OH}/k_{CD3OD}$ (~5.2, Fig. 5b), predicating the $CH_3OH$ activation on the nanoscaled Pt particles and the negligible effects of the $CeO_2$ supports. Comparatively, the significantly decreased $k_{CH3OH}/k_{CD3OD}$

value (2.6) for $Pt_1/PN\text{-}CeO_2$ indicated the single-atom Pt benefits the activation of $CH_3OH$ (Fig. 5b). Therefore, isotope experiments strongly suggest that the dual-active sites of single-atom Pt and FLPs are kinetically beneficial for the simultaneously promoted activation of $CH_3OH$.

In comparison with $Pt/PN\text{-}CeO_2$ and $Pt/NR\text{-}CeO_2$, the $Pt_1/PN\text{-}CeO_2$ catalysts exhibited the highest $H_2$ generation rate at each reaction temperature (Fig. 5c). More importantly, the ratios of $H_2$ generation rates of $Pt_1/PN\text{-}CeO_2$ to $Pt/PN\text{-}CeO_2$ increased gradually from 1.79, to 2.22, and then to 2.37, and final to 3.75 at reaction temperatures of 180, 165, 150, and 135 °C, respectively (Fig. 5c), demonstrating the highest capability of $Pt_1/PN\text{-}CeO_2$ for $H_2$ generation, especially at low temperatures. Also, the derived activation energy ($E_a$) of $Pt_1/PN\text{-}CeO_2$ for $H_2$ generation via APMR reaction was 70.8 kJ mol$^{-1}$, which was 1.4 and 1.7 times lower than the $E_a$ of $Pt/PN\text{-}CeO_2$ (95.3 kJ mol$^{-1}$) and $Pt/NR\text{-}CeO_2$ (120.0 kJ mol$^{-1}$), respectively (Fig. 5d and Supplementary Fig. 20). Therefore, the improved activation of $H_2O$ and $CH_3OH$ results in the decreased energy barrier of $H_2$ generation, further leading to the remarkable catalytic performance of $Pt_1/PN\text{-}CeO_2$ at low temperatures.

Experimentally, the single-atom Pt sites for the effective $CH_3OH$ activation have been proved by the obviously decreased value of $k_{CH3OH}/k_{CD3OD}$ for $Pt_1/PN\text{-}CeO_2$. In order to gain deep insights, DFT calculations were also employed to understand the activation of $CH_3OH$ on the constructed $Pt_1$-FLP dual-active sites (Fig. 1d). Three catalytic models of Pt (111), $Pt_1/CeO_2(110)$, and $Pt_1/CeO_2(110)$-FLP were also used to represent the Pt nanoparticles, single-atom Pt on oxygen vacancy of $CeO_2(110)$ and single-atom Pt on FLP site of $CeO_2(110)$, respectively. As shown in Supplementary Fig. 21, the largest energy barrier of 0.75 eV for $CH_3OH$ decomposition on Pt(111), is the first step of $CH_3OH$ transformation into *$CH_3O$ and *H, consistent with literature[11]. For $Pt_1/CeO_2(110)$, the step of *$CH_3O$ transformation into *$CH_2O$ and *H exhibits a similar energy barrier (0.8 eV) with it on Pt(111) surface. These results revealed that the size of Pt was not the critical

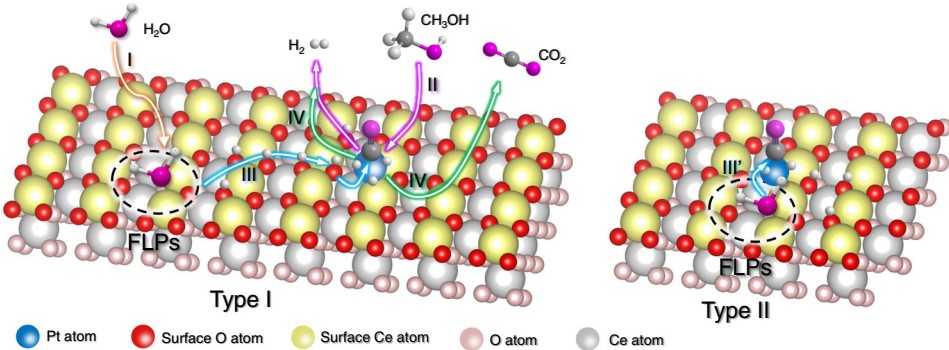

**Fig. 6 | Proposed reaction process.** Catalytic pathway for $H_2$ generation from methanol and $H_2O$ catalyzed by the $Pt_1$-FLP dual-active sites constructed on $Pt_1$/PN-CeO$_2$.

factor for $CH_3OH$ dissociation. However, the energy barrier for the $CH_3OH$ decomposition is significantly reduced to only 0.23 eV on $Pt_1$/$CeO_2$(110)-FLP along with the transformation of *CHO into *CO and *H (Supplementary Fig. 21). In this configuration, the Lewis acidic $Ce^{3+}$ as a co-active site interacts with $CH_3OH$ molecule, further promoting the $CH_3OH$ dissociation afterward. Therefore, the presence of FLP sites adjacent to single-atom Pt results in the obvious reduced energy barrier of $CH_3OH$ dissociation.

To experimentally detect the roles of Lewis acidic $Ce^{3+}$, the adsorption behavior of methanol on the surface of $Pt_1$/PN-CeO$_2$ catalysts were investigated. As shown in Supplementary Fig. 22, methanol molecules could adsorb on the oxygen vacancy of PN-CeO$_2$ and then dissociate to methoxy species owing to the presence of the bridged methoxy species with the characteristic peak at 1033 cm$^{-1}$. After introducing single-atom Pt, another bridged methoxy species at 1058 cm$^{-1}$ appeared, as revealed from the FTIR spectra. Compared with the adsorption behavior of methanol on PN-CeO$_2$, this appeared methoxy species could be attributed to the bridged configuration of methoxy on the single-atom Pt and adjacent Ce[50]. Previous reports have proved that the $CeO_2$ supports exhibited no catalytic ability for methanol dissociation[27]. Therefore, the bridged methoxy species on the single-atom Pt and adjacent Ce were detected as the critical intermediate for methanol dissociation on the $Pt_1$/PN-CeO$_2$ catalysts.

In addition, the d-band electron structures of various Pt sites were characterized by high-resolution valence-band XPS spectra. Due to the synergistic effects of the PN-CeO$_2$ supports with abundant defect of oxygen vacancy and the atomic dispersion of Pt active sites, the $Pt_1$/PN-CeO$_2$ catalysts exhibited a significant shift of the d-band center towards the VBM (Supplementary Fig. 23), resulting in a downward shift of the antibonding states. According to the d-band center theory, the strong bonding of adsorbates occurs if the antibonding states are shifted up relative to the Fermi level, and weak bonding occurs if the antibonding states are shifted down[51,52]. Therefore, the $Pt_1$/PN-CeO$_2$ catalysts also exhibited the weakest bonding with $CO_2$ molecule, facilitating its desorption form single-atom Pt active sites.

Based on the various control experiments, kinetic analysis, and DFT calculations, the constructed $Pt_1$-FLP dual-active site in $Pt_1$/PN-CeO$_2$ boosted a high catalytic performance for $H_2$ generation through APMR at low temperatures. A proposed catalytic process is illustrated in Fig. 6. **(I)** $H_2O$ molecule is adsorbed and dissociated into *H and *OH species on the FLP sites. **(II)** Methanol molecule is easily dissociated into *H and *CO on the single-atom Pt with the help of the Lewis acidic $Ce^{3+}$ site of FLP sites or the nearly $Ce^{3+}$ site, releasing $H_2$ molecule in methanol. For the type **I** of $Pt_1$-FLP dual-active sites (Fig. 6a), due to the long spatial distance between Pt and FLPs, **(III)** the *H and *OH species would diffuse on the surface of $CeO_2$ from FLP sites to the single-atom Pt sites, and then reform of *CO intermediates on the interfacial Pt active sites to give $CO_2$ along with the production of extra $H_2$. **(IV)** The generated $H_2$ and $CO_2$ molecules release from the surface of catalysts. Meanwhile, for the type **II** of $Pt_1$-FLP dual-active sites (Fig. 6b), **(III)**, with the help of the

abundant *OH species on FLP sites, the reforming of *CO intermediate effectively occurs on the adjacent Pt active sites without the diffusion of the *H and *OH species. Other steps on the type **II** of $Pt_1$-FLP dual-active sites are similar to those on the type **I** of $Pt_1$-FLP dual-active sites. Therefore, the $Pt_1$/PN-CeO$_2$ catalysts with the richest interface between Pt and PN-CeO$_2$ and abundant FLP sites enable the efficient $CH_3OH$ and $H_2O$ dissociation and effectively reformed the *OH and *CO intermediate, facilitating the $H_2$ production at low temperatures.

## Discussion

The dual-active site catalyst that comprises the atomically dispersed Pt and FLPs on PN-CeO$_2$ has been successfully developed for the low-temperature $H_2$ generation with the suppressed CO from methanol and $H_2O$. Methanol is efficiently dissociated into *H and *CO intermediates with the help of Lewis acidic $Ce^{3+}$ sites of FLPs, in which the energy barrier is only 0.23 eV. The FLP sites constructed on PN-CeO$_2$ enable the kinetically and dynamically favorable $H_2O$ dissociation, producing abundant surface hydroxyls for the subsequent transformation of *CO and *OH into *CO$_2$ and *H. Therefore, the $H_2$ generation from methanol and $H_2O$ at low temperatures is significantly accelerated between single-atom Pt and FLP sites. Due to hydrogen with the low CO concentration required of fuel cells as well as other applications, herein, the catalytically generated hydrogen with a CO concentration of 270 ppm from methanol and $H_2O$ by $Pt_1$/PN-CeO$_2$ has to be handled carefully and purified as the power supplies of the fuel cell at this stage. Nevertheless, this new catalyst featured with the facile synthesis and high activity as well as the suppressed CO generation at low temperatures still paves a possible way towards a commercially achievable liquid sunshine roadmap.

## Methods
### Preparation of PN-CeO$_2$
The PN-CeO$_2$ supports were prepared by a two-step hydrothermal process at various temperatures[47,53]. Initially, aqueous solutions of Ce(NO$_3$)$_3$·6H$_2$O (1.736 g in 10 mL of deionized water) and NaOH (19.2 g in 70 mL of deionized water) were mixed slowly in a Pyrex bottle (100 mL) and reacted for 0.5 h at room temperature under continuous stirring. After aging for another 1 h, the reaction was continued at 100 °C for 24 h for the first hydrothermal process. Then, the reaction mixture was cooled naturally to room temperature. The CeO$_2$/Ce(OH)$_3$ solids were collected by centrifugation, intermittently washed with deionized water and ethanol for three times, and dried in air at 60 °C. After that, the secondary hydrothermal process was performed to treat the CeO$_2$/Ce(OH)$_3$ precursors with 2 mg mL$^{-1}$ at 100 °C for 12 h. Finally, the PN-CeO$_2$ supports were obtained after centrifugation and dried in air at 60 °C.

### Preparation of the $Pt_1$/PN-CeO$_2$ catalysts
The $Pt_1$/PN-CeO$_2$ catalysts were synthesized through a photo-assisted deposition process. Initially, 300 mg of PN-CeO$_2$ supports were

dispersed in 40 mL of 6 vol.% methanol aqueous solution. After adding the desired amount of $H_2PtCl_6$, the mixture was irradiated under Xe lighter for 3 h. Then, the products were collected by centrifugation to remove the free $H_2PtCl_6$. Finally, the $Pt_1/PN-CeO_2$ catalysts were reduced by 5 vol.% $H_2/Ar$ at 300 °C for 2 h.

## Characterizations

The catalysts were characterized by a Shimadzu X-ray diffractometer (Model 6000) using Cu Kα radiation. TEM studies were conducted on the Hitachi HT-7700 with an accelerating voltage of 120 kV. High-resolution and dark-field TEM images were acquired from the Tecnai G2 F20 S-twin transmission electron microscope at 200 kV. The surface area was measured by $N_2$ physisorption (Micromeritics, ASAP 2020 HD88) based on Brunauer-Emmet-Teller (BET) method. XPS were acquired using a Thermo Electron Model K-Alpha with Al $K_\alpha$ as the excitation source.

## Catalytic performance for hydrogen generation from methanol and $H_2O$

For a typical catalytic reaction, 40 mL of methanol and 18 mL of $H_2O$ with 50 mg catalysts were mixed in a 500 mL autoclave equipped with a temperature controller and a pressure detector. The temperature of the reaction system quickly increased to the given temperature within 10 min. After the reaction, the mixture was collected in a 1 L of gas sampling bag and then analyzed by GC with TCD and FID detector.

## Data availability

The authors declare that the main data supporting the findings of this are available within the article and supplementary information from the corresponding author upon reasonable request. The main data generated in this study are provided in the Source data file. Source data are provided with this paper.

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

## Acknowledgements

We acknowledge the National Natural Science Foundation of China (21872109 Y.Q. and 22002115 S.Z.). S.Z. is supported by the Guangdong Basic and Applied Basic Research Foundation (2022B1515020092) and the Young Elite Scientists Sponsorship Program by CAST (2019QNRC001). We also acknowledge the Fundamental Research Funds for the Central Universities (D5000210283 S.Z., D5000210601 Y.Y., and D5000210829 Y.Q.). This research project was supported by TianHe-2 at Shanxi Supercomputing Center of China and Central for High Performance Computing of Northwestern Polytechnical University, China. We gratefully acknowledge XAS measurements at the BL14W1 beamline of the Shanghai Synchrotron Radiation Facility.

## Author contributions

S.Z. and Y.Q. designed the studies and wrote the paper. S.Z., Y.L., and M.Z. performed most of the experiments. J.H. carried out the DFT calculations. S.Z. and Y.Q. performed data analysis. All authors discussed the results and commented on the manuscript.

## Competing interests

The authors declare no competing interests.
