## [Peer Review File · Nature Communications]

Title: Sustainable production of hydrogen with high purity from methanol and water at low temperaturesREVIEWER COMMENTS

Reviewer #1 (Remarks to the Author):

The paper by Zhang et al. tackles an interesting subject, namely low temperature methanol water reforming over a catalyst of dual active sites of Pt single-atoms and frustrated Lewis pairs (FLPs) on porous nanorods of CeO₂. This reviewer ranks the experimental catalysis and the DFT calculation in this paper as important for the understanding of the activation of H₂O and methanol at low temperature. However, the role of the frustrated Lewis pairs (FLPs), the synergy of single Pt and FLPs in this study were not demonstrated and discussed sufficiently. This reviewer supports the publication in Nature Comm., but only after addressing the open points in the results analysis section in an appropriate manner:

Firstly, there are some confusions, this reviewer would like to discuss with the authors. It is about the key words of “Sustainable production of hydrogen, high purity, methanol and water at low temperatures” in the title.

- Page 10, line 187-189 “To the best of our knowledge, it is the lowest temperature for all previously reported heterogeneous catalysts to achieve the efficient H₂ generation of reforming of methanol and water in the absence of any additives.” This conclusion is inaccurate: please check the study of ACS Appl. Mater. Interfaces 2021, 13, 24702–24709
- In abstract section: (page two, line 19-22) “Herein, we demonstrate that the dual-active sites of Pt single-atoms and frustrated Lewis pairs (FLPs) on porous nanorods of CeO₂ enable the efficient additive-free H₂ generation with a low CO generation (0.027 %) through APRM at 120 °C.” As we know, CO is a poison for fuel cell, and normally, even miniscule amounts of carbon monoxide (50-100 ppm) in the hydrogen are sufficient to bind to the platinum catalysts and prevent them working. Here, the 0.027 % is 270 ppm of CO. The author should discuss how to handle such high CO concentration in real application case.
- In table S1, with the same single Pt catalyst, H₂ generation rate (molH₂ molPt⁻¹ h⁻¹) at 135 °C is 199, while, at 120 °C it is decreased to 33. What are the advantages of carrying out this reaction with 15 °C difference, but losing 83.4% activity [(1-33/199)%]? In page 10, line 192-193 “When the reaction temperature was further increased to 165 °C, the H₂ generation rate was significantly enhanced to 1103 molH₂ molPt⁻¹ h⁻¹.” The activity losing with 165 °C as a standard is about 97 % [(1-33/1103)%]. It seems that a lower temperature (100-135 °C) is not in the optimal working range, why the authors emphasize the advantages of low temperature? This reviewer would suggest the authors, to compare your catalyst and the catalyst in Nature, 2017, 544, 80-83 (Reference 11 in manuscript) at a wider reaction temperature range.
- The last one, the authors should further clarify the concept of “Sustainable production of hydrogen” with methanol as H₂ carrier. Industry methanol is produced from synthesis gas (carbon monoxide and

hydrogen). And one product of methanol water reforming is CO₂.

Secondly, this study provides a systematic experiment and demonstrates a unique catalytic performance of loading single Pt atom on the surface of CeO_x. There are some comments this reviewer would like to share with the authors:

- Single Pt/CeO_x catalyst has been widely studied. The single metal Pt-CeO_x support interaction is quite important for its catalytic performance. The authors introduce a concept of “the dual-active sites of Pt single-atoms and frustrated Lewis pairs (FLPs)” to understand the surface chemistry. This reviewer consider of this concept is the major innovation in this study. This reviewer suggests the authors to provide more evidence to further clarify: the 1) coordination configuration of single Pt; 2) geometry model of FLPs, (Figure 1D is hard to understand); 3) the geometry model of single Pt with FLPs; 4) a detailed reaction pathway on this Single Pt-FLPs (reedit Figure 6). In the reaction mechanism model, is there any synergy between the single Pt with FLPs. And the distance between single Pt and FLPs matter?
- HAADF-STEM image and XANES spectra can not rule out the possibility of the existence of PtO cluster. This reviewer suggests a CO-FTIR analysis. (Single Pt on CeO_x has been reported to have a very low binding energy with CO)
- H/D isotope experiments and DFT calculations are well done. However, to further understand the reaction pathway of methanol-water reforming at low temperature, the authors should provide in-situ spectroscopic evidence, to detect/monitor the reaction intermediates.

The dynamic structure of active sites in ceria supported Pt catalysts is very interesting research topic. Here, this reviewer suggests the authors to discuss the study of (Li, Y., Kottwitz, M., Vincent, J.L. et al. Dynamic structure of active sites in ceria-supported Pt catalysts for the water gas shift reaction. Nat Commun 12, 914 (2021).).

Reviewer #2 (Remarks to the Author):

In this present work, Sai Zhang et al. synthesized Pt single atom deposited porous CeO₂ nanorods to produce H₂ from a water and methanol mixture. The frustrated Lewis pairs of porous nanorods of CeO₂ play a vital role in methanol activation along with Pt single atoms. However, the concept of frustrated Lewis pairs is not clear and experimentally unresolved. It is tough to recommend this paper for publication at this stage.

Comments to the authors

1. The concept of frustrated Lewis pairs is not clear, and experimental evidence is not enough to support it. Since frustrated Lewis pairs is the heart of the work, it should be detailed with some experimental evidence.
2. The d-band center of Pt atoms is one of the main factors for CO intermediate adsorption (poisoning). The d-band position should be calculated for bare Pt, Pt single atom, and Pt single atom-CeO₂ nanorods catalyst by UPS analysis.
3. DFT calculations can also be done for a d-band center shift concerning the Fermi level.

4. At least CO stripping experiment should be performed by cyclic voltammetry measurement for all the synthesized catalysts.
5. The surface area and porosity of CeO₂ nanorods and Pt single atom nanorods CeO₂ catalyst should be disclosed by BET analysis.
6. The wt % of Pt atoms should be ascertained by ICP-OES measurements.
7. The frustrated Lewis pairs in the PN-CeO₂, NR- CeO₂, and NP-CeO₂ catalyst after loading of Pt is not studied. So, it is hard to understand whether Pt influences the frustrated Lewis pairs formation or not.
8. Why does a 20 % loss of catalytic performance after the stability test (Figure 3C)? Is Pt single atom leaching out from CeO₂ nanorods?
9. Besides, after the catalytic reactions, the existence of frustrated Lewis pairs sites should be studied.

Reviewer #1 (Remarks to the Author):

The paper by Zhang et al. tackles an interesting subject, namely low temperature methanol water reforming over a catalyst of dual dual-active sites of Pt single-atoms and frustrated Lewis pairs (FLPs) on porous nanorods of CeO₂. This reviewer ranks the experimental catalysis and the DFT calculation in this paper as important for the understanding of the activation of H₂O and methanol at low temperature. However, the role of the frustrated Lewis pairs (FLPs), the synergy of single Pt and FLPs in this study were not demonstrated and discussed sufficiently. This reviewer supports the publication in Nature Comm., but only after addressing the open points in the results analysis section in an appropriate manner:

Response: Thank for the positive and constructive comments raised by the reviewer. We would like to address those comments in details.

Firstly, there are some confusions, this reviewer would like to discuss with the authors. It is about the key works of “Sustainable production of hydrogen, high purity, methanol and water at low temperatures” in the title.

- Page 10, line 187-189 “To the best of our knowledge, it is the lowest temperature for all previously reported heterogeneous catalysts to achieve the efficient H₂ generation of reforming of methanol and water in the absence of any additives.” This conclusion is inaccurate: please check the study of ACS Appl. Mater. Interfaces 2021, 13, 24702–24709

Response: Thank you for your kind reminder. The H₂ production of reforming of methanol and water has been successfully achieved by the N-doped carbon dots/g-C₃N₄ catalysts in the study of ACS Appl. Mater. Interfaces 2021, 13, 24702-24709. The carbon dots provide a catalytic hydrogen generation from methanol and H₂O without transition metal catalysts, which is of great significant in this field. In that study, the H₂ generation rate was 19.5 μmol g⁻¹ h⁻¹, which is much lower than the rate catalyzed by Pt₁/PN-CeO₂ (6400 μmol g⁻¹ h⁻¹) in this work. Nonetheless, our judgment is inaccurate in the initial submission. We have summarized the relevant data in the Table S3 to facilitate readers to grasp the research progress more comprehensively.

Meanwhile, we have modified the sentence to “*To the best of our knowledge, it is among one of the lowest temperatures for all previously reported heterogeneous metal catalysts to achieve the efficient H₂ generation of*”

reforming of methanol and water in the absence of any additives.” (Page 11, Line 215-218)

• In abstract section: (page two, line 19-22) “Herein, we demonstrate that the dual-active sites of Pt single-atoms and frustrated Lewis pairs (FLPs) on porous nanorods of CeO₂ enable the efficient additive-free H₂ generation with a low CO generation (0.027 %) through APRM at 120 °C.” As we know, CO is a poison for fuel cell, and normally, even miniscule amounts of carbon monoxide (50-100 ppm) in the hydrogen are sufficient to bind to the platinum catalysts and prevent them working. Here, the 0.027 % is 270 ppm of CO. The author should discuss how to handle such high CO concentration in real application case.

Response: This is a practical question for the commercial application of liquid methanol as hydrogen storage. Methanol and H₂O system offers a promising methodology for transportation and storage. In this study, the selectivity of CO is among one of the lowest levels in comparison with those of the previous studies. However, the generated H₂ from this system has to be purified before it supplies the power of fuel cell at this stage.

We have added the relevant discussion in the revised manuscript, as following:

“Due to hydrogen with the low CO concentration required of fuel cells as well as other applications, herein, the catalytically generated hydrogen with a CO concentration of 270 ppm from methanol and H₂O by Pt₁/PN-CeO₂ has to be handled carefully and purified as the power supplies of fuel cell at this stage. Nevertheless, this new catalyst featured with the facile synthesis and high activity as well as the suppressed CO generation at low temperatures still paves a possible way towards a commercially achievable liquid sunshine roadmap.” (Line 440-446, Page 21)

• In table S1, with the same single Pt catalyst, H₂ generation rate (mol_{H₂} mol_{Pt}⁻¹ h⁻¹) at 135 °C is 199, while, at 120 °C it is decreased to 33. What are the advantages of carrying out this reaction with 15 °C difference, but losing 83.4% activity [(1-33/199)%]? In page 10, line 192-193 “When the reaction temperature was further increased to 165 °C, the H₂ generation rate was significantly enhanced to 1103 mol_{H₂} mol_{Pt}⁻¹ h⁻¹.” The activity losing with 165 °C as a standard is about 97 % [(1-33/1103)%]. It seems that a lower temperature (100-135 °C) is not in the optimal working range, why the authors emphasize the advantages of low temperature? This reviewer would suggest the authors, to compare your catalyst and the catalyst in Nature, 2017, 544, 80-83 (Reference 11 in manuscript) at a wider reaction temperature range.

Response: Thank you for your comments. Up to now, the reforming of methanol and water by those catalysts

still faces two big obstacles: (1) the high temperatures ($> 250\text{ }^{\circ}\text{C}$) to boost catalytic reaction and (2) the low purity of H_2 accompanied with the generation of CO at a high level. Recent advances in developing new heterogeneous catalysts have greatly decreased the operation temperatures as low as $150\text{ }^{\circ}\text{C}$ for the aqueous phase reforming of methanol by using the atomically dispersed Pt on $\alpha\text{-MoC}$. Afterwards, the further decrease of the reaction temperatures to yield a satisfactory hydrogen generation rate is extremely difficult and rarely realized on heterogeneous catalysts yet up to now. Therefore, developing high-efficient catalysts capable of *in-situ* releasing of H_2 at even lower temperatures and the suppressed CO generation is highly desirable for the large-scale production of hydrogen, bringing us a step closer to methanol economy.

Herein, we demonstrate that the dual-active site catalysts composed of the single-atom Pt and frustrated Lewis pairs (FLPs) on the atomically dispersed Pt anchored on porous nanorods of CeO_2 ($\text{Pt}_1/\text{PN-CeO}_2$) enables a stabilized H_2 generation at a low-temperature of $120\text{ }^{\circ}\text{C}$ through a base-free aqueous phase reforming of methanol. However, the hydrogen generation rate ($1592\text{ mol}_{\text{H}_2}\text{ mol}_{\text{Pt}}^{-1}\text{ h}^{-1}$) catalyzed by $\text{Pt}_1/\text{PN-CeO}_2$ at $180\text{ }^{\circ}\text{C}$ is lower than the rate catalyzed by 2% Pt/ $\alpha\text{-MoC}$ at $170\text{ }^{\circ}\text{C}$ ($1755\text{ mol}_{\text{H}_2}\text{ mol}_{\text{Pt}}^{-1}\text{ h}^{-1}$) in the reference (Nature, 2017, 544, 80-83). We did not perform the hydrogen generation at a higher temperature owing to the pressure limitation of our autoclave. The main contribution of our manuscript is the satisfactory hydrogen generation with a low CO generation from methanol and H_2O at low temperature ($120\text{ }^{\circ}\text{C}$ and even $100\text{ }^{\circ}\text{C}$) *via* the novel dual-active sites of single-atom Pt and FLPs sites for methanol and H_2O activation, respectively. Therefore, this new catalyst featured with facile synthesis, high activity and suppressed CO generation at low temperature paves the way towards a commercially achievable liquid sunshine roadmap.

- The last one, the authors should further clarify the concept of “Sustainable production of hydrogen” with methanol as H_2 carrier. Industry methanol is produced from synthesis gas (carbon monoxide and hydrogen). And one product of methanol water reforming is CO_2 .

Response: Undoubtedly, industry methanol is produced from synthesis gas. However, from the perspective of technological development, the sustainable production of hydrogen is achieved through the following steps: **(I)** As a promising methodology for hydrogen storage *via* methanol, the H_2 source must be green hydrogen, which is transformed from solar, wind and/or other renewable energy. **(II)** After that, green methanol can be generated by the catalytic reaction between CO_2 and H_2 . Also, the green methanol can be obtained from transformation of biomass.

(III) Finally, hydrogen production can be achieved *via* reforming of green methanol and H₂O. (IV) After reforming, generated CO₂ can be further transformed into methanol again *via* hydrogenation of CO₂ by green H₂. Therefore, sustainable production of hydrogen is achieved through green methanol and CO₂ recycling. This concept has been reported and summarized in a previous review by Prof. Choon FongShih, Prof. Tao Zhang, Prof. Jinghai Li, and Prof. Chunli Bai (*Joule*, 2018, 2, 1925-1949). Thus, it is fine to clarify the “Sustainable production of hydrogen”.

Secondly, this study provides a systematic experiment and demonstrates a unique catalytic performance of loading single Pt atom on the surface of CeO_x. There are some comments this reviewer would like to share with the authors:

- Single Pt/CeO_x catalyst has been widely studied. The single metal Pt-CeO_x support interaction is quite important for its catalytic performance. The authors introduce a concept of “the dual-active sites of Pt single-atoms and frustrated Lewis pairs (FLPs)” to understand the surface chemistry. This reviewer consider of this concept is the major innovation in this study. This reviewer suggests the authors to provide more evidence to further clarify: the 1) coordination configuration of single Pt; 2) geometry model of FLPs, (Figure 1D is hard to understand); 3) the geometry model of single Pt with FLPs; 4) a detailed reaction pathway on this Single Pt-FLPs (reedit Figure 6). In the reaction mechanism model, is there any synergy between the single Pt with FLPs. And the distance between single Pt and FLPs matter?

Response: Thank you for your constructive comments. We would like to answer these questions in the following order:

1) **Geometry model of FLPs:**

The geometry model and construction process have been systematically discussed in our previous reports (*Nat. Commun.* 2017, 8, 15266; *Chem. Soc. Rev.* 2018, 47, 5541-5553; *J. Am. Chem. Soc.* 2019, 141, 11353-11357). To make it easier for the reader to understand, the geometry model of FLPs was added in our revised manuscript. As shown in Figure S1, the FLPs site is constructed by the two adjacent surface Ce³⁺ as Lewis acidic site and the neighboring surface lattice oxygen as Lewis basic site on CeO₂(110) surface.

Figure S1. (a) Optimized structure and (b) electron-density isosurface of FLPs sites on CeO₂(110) surface.

The constructive process of FLP site has been also described in the revised manuscript. As shown in Figure 2a, for ideal CeO₂(110) surface, surface Ce and O atoms form the typical Lewis acid-base adjuncts. The surface oxygen defect is easily formed on CeO₂ surface. When one O atom (O_{Ib}) is removed from CeO₂(110) surface, the Lewis acidic Ce_I and Lewis basic O_{IIc} atoms are expected to construct FLP sites of Ce_I-O_{IIc} after structural relaxation (Figure 2b). However, the electronic interaction between Ce_I and O_{Ia}/O_{Ic} will still hinder the activation of small molecules on the Ce_I-O_{IIc} active sites (Figure S2e). Therefore, the weak FLPs-like activation is obtained on the traditional CeO₂ materials. When the second adjacent surface oxygen (O_{Ia}) is removed, the reduced Ce cations (Ce_I and Ce_{II}) and surface lattice oxygen (O_{IIc}) are independent Lewis acidic and basis sites (Figure S2c and S2f), respectively. Notably, two adjacent reduced surface Ce sites (Ce_I and Ce_{II}) and lattice O_{IIc} is constructed the FLPs site of (Ce_I,Ce_{II})-O_{IIc} with a shorter distance (3.99 Å).

However, the construction of surface FLPs site cannot be realized by removing surface oxygen atoms on the CeO₂(111) facet. Meanwhile, due to the low formation of oxygen vacancy on CeO₂(100) surface, the relatively unstable oxygen defect leads to the spatial configuration for formation of FLPs sites. Therefore, CeO₂(110) surface instead of CeO₂(100) and CeO₂(111) surfaces exhibits the highest possibility for FLPs construction owing to the unique FLPs configuration and formation energy of oxygen vacancies on various surfaces.

We have added this relevant description in the revised manuscript. (Supporting Information, Figure S1 and S2)

Figure S2. Schematic images of concept for design of solid frustrated Lewis pairs in CeO₂ crystal structure. (a) Optimized structure of ideal CeO₂(110). (b) Optimized structure of CeO₂(110) with one oxygen vacancy. (c) Optimized structure of CeO₂(110) with two adjacent oxygen vacancies. (d) Electron-density isosurface of ideal CeO₂(110). (e) Electron-density isosurface of CeO₂(110) with one oxygen vacancy. (f) Electron-density isosurface of CeO₂(110) with two oxygen vacancies. The electron-density isosurfaces are plotted at 0.01 e/bohr³. The color bar represents the electrostatic potential scale.

2) Coordination configuration of single Pt.

According to the aberration-corrected high-angle annular dark field scanning transmission electron microscopy (HAADF-STEM) image (Figure 2a), the single-atom Pt in catalysts were experimentally demonstrated, which was further verified from the uniform Pt distribution on PN-CeO₂ by the energy dispersive spectroscopy (EDS) mapping (Figure 2b). X-ray absorption near edge structures (XANES) of Pt K-edge revealed that the white line peak of the Pt₁/PN-CeO₂ catalysts located at 11568.7 eV (Figure 2c), which was very close to that of PtO₂. More importantly, the *k*³-weight Fourier transforms of extended X-ray absorption fine structure (EXAFS) spectra of Pt₁/PN-CeO₂ delivered one prominent peak at ~1.63 Å, which was labeled as Pt-O bond (Figure 2d). Therefore, the single-atom Pt was undoubtedly coordinated with O atom on CeO₂ surface.

Meanwhile, DFT simulation was used to further explore the coordination configuration of single Pt. As shown in Figure S3, the most stable configuration is the Pt atom located in oxygen vacancy of CeO₂(110) surface with the lowest adsorption energy. Meanwhile, single Pt atom can be stabilized on oxygen vacancy adjacent to the FLPs sites of CeO₂(110). On this configuration, the oxygen vacancies adjacent to the FLPs sites is occupied by a single Pt atom. Consistent with the experimental results, the single Pt atom coordinates with different number of O atoms in these

two typical configurations. Therefore, the single atom Pt is located on the oxygen vacancy and coordinated with O atom on CeO₂ surface.

Figure S3. Summary of the adsorption energy of Pt single-atom at various sites on CeO₂(110) surface.

3) *The geometry model of single Pt with FLPs:*

Response: Considering that the single atom Pt active sites exhibit higher capability for CH₃OH activation than Pt nanoparticles, the constructed dual-active sites of single-atom Pt and FLPs (Pt₁-FLP) can effectively activate both H₂O and CH₃OH, potentially reducing the reaction temperatures of APRM to generate H₂. DFT calculations were also used to explore the possible spatial structure of Pt atom on CeO₂(110) surface. As shown in Figure S3, the single-atom Pt is easier to form on the oxygen defect of CeO₂(110) surface owing to the lowest formation energy (1.35 eV). In this spatial configuration, the dual-active sites of single-atom Pt and FLPs sites is successfully designed, as type **I** in Figure 1d. where Pt atom is located on the oxygen vacancy of CeO₂(110) surface and in the distance with FLPs sites. In addition, the single-atom Pt could occupy one of the oxygen vacancies adjacent to the FLPs sites owing to the slightly high formation energy (1.70 eV). For this spatial configuration, the Pt₁-FLP dual-active sites is spatially adjacent with each other, as type **II** in Figure 1d. Fortunately, the spatial configuration of FLPs site, which is constructed by two adjacent Ce³⁺ as Lewis acidic site and lattice O atom as Lewis basic site, is not affected by the nearby single Pt atom as well as single Pt atom in the distance of the two configurations.

Figure 1d. The designed dual-active sites of single-atom Pt and FLPs. Type **I** is the dual-active sites of single-atom Pt and FLPs site in the distance. Type **II** is the dual-active sites of adjacent single-atom Pt and FLPs site. Note: The dark blue, red and yellow balls respective the Pt, O and Ce atoms, respectively.

Meanwhile, the dispersion and chemical environment of Pt on PN-CeO₂ was studied by diffuse-reflectance infrared Fourier-transform spectroscopy (DRIFTS). The peak at ~ 2090 cm⁻¹ was assigned to the linearly adsorbed CO on the isolated ionic Pt²⁺ (Figure S8), revealing that the Pt active sites existed as single-atom on PN-CeO₂ and coordinated with O atoms.¹⁻³ Specifically, the two apparent peaks at 2099 cm⁻¹ and 2076 cm⁻¹ could be attributed to the single-atom Pt in spatial configuration of type **I** and type **II** (Figure 1d), respectively, owing to the lower valence state of Pt on oxygen vacancy adjacent to the FLPs sites than it on oxygen vacancy in the distance (Figure S3). Therefore, combining with the HAADF-STEM, XANES and DRIFTS results, the dual-active sites of single-atom Pt and FLPs were successfully constructed on the surface of Pt₁/PN-CeO₂ catalysts, where the single-atom Pt was mainly located in oxygen vacancy.

Figure S8. The DRIFTS analysis of the CO adsorption on the surface of Pt₁/PN-CeO₂.

We have modified and supplemented relevant description in the revised manuscript, as following:

“Then, DFT calculations were further used to explore the possible spatial structures of Pt atom on CeO₂(110) surface, which delivered two configurations. As shown in Figure S3, the Pt atom prefers to occupy the oxygen defect of CeO₂(110) surface owing to the lowest formation energy (1.35 eV). In this configuration, the FLP site is not affected by the single-atom Pt in the distance. Therefore, the Type I of Pt₁-FLP dual-active site is successfully constructed, in which Pt single-atom locates at the oxygen vacancy of CeO₂(110) surface (Figure 1d). In addition, the Pt single-atom can occupy one of the oxygen vacancies adjacent to the FLP site with a slightly high formation energy (1.70 eV). For this configuration, the type II of Pt₁-FLP dual-active site is spatially adjacent to each other, as shown in Figure 1d and S3. Also, the spatial and electronic configuration of FLP site are preserved in the presence of the nearby single Pt atom. (Page 7, Line 129-140)

.....

Then, the dispersion and chemical environments of Pt on PN-CeO₂ were studied by diffuse-reflectance infrared Fourier-transform spectroscopy (DRIFTS). The peak at ~ 2090 cm⁻¹ was assigned to the linearly adsorbed CO on isolated ionic Pt²⁺ (Figure S8),¹⁻³ revealing that the Pt active sites existed as atomic dispersion on the surface of PN-CeO₂ and coordinated with O atoms. Specifically, the two apparent peaks at 2099 cm⁻¹ and 2076 cm⁻¹ could be attributed to the single-atom Pt located in the configurations of the type I and type II (Figure 1d), respectively, owing to the lower valence state of Pt on oxygen vacancy adjacent to the FLP sites than it on oxygen vacancy in the distance (Figure S3). Therefore, combining with the HAADF-STEM, XANES and DRIFTS results, the dual-active sites of single-atom Pt and FLPs were successfully constructed on the surface of Pt₁/PN-CeO₂ catalysts, where the single-atom Pt mainly occupied oxygen vacancy.” (Page 10, Line 185-196)

4) A detailed reaction pathway on this Single Pt-FLPs:

Response: The detailed reaction pathway has been modified as following:

“Based on the various control experiments, kinetic analysis and DFT calculations, the constructed Pt₁-FLP dual-active site in Pt₁/PN-CeO₂ boosted a high catalytic performance for H₂ generation through APMR at low temperatures. A proposed catalytic process is illustrated in Figure 6. (I) H₂O molecule is adsorbed and dissociated into *H and *OH species on the FLP sites. (II) Methanol molecule is easily dissociated into *H and *CO on the single-atom Pt with the help of the Lewis acidic Ce³⁺ site of FLP sites or the nearby Ce³⁺ site, releasing of H₂ molecule in methanol. For the type I of Pt₁-FLP dual-active sites (Figure 6a), due to the long spatial distance between Pt and FLPs, (III) the *H and *OH species would diffuse on the surface of CeO₂ from FLP sites to the single-atom Pt sites, and then reform of *CO intermediates on the interfacial Pt active sites to give CO₂ along with the production of extra H₂. (IV) The generated H₂ and CO₂ molecules release from the surface of catalysts. Meanwhile, for the type II of Pt₁-FLP dual-active sites (Figure 6b), (III') with the help of the abundant *OH species on FLP sites, the reforming of *CO intermediate effectively occurs on the adjacent Pt active sites without the diffusion of the *H and *OH species. Other steps on the type II of Pt₁-FLP dual-active sites are similar to those on the type I of Pt₁-FLP dual-active sites. Therefore, the Pt₁/PN-CeO₂ catalysts with the richest interface between Pt and PN-CeO₂ and abundant FLP sites enable the efficient CH₃OH and H₂O dissociation and effectively reformed of the *OH and *CO intermediate, facilitating the H₂ production at low temperatures.” (Page 20, Line 411-430)

Figure 6 | Proposed reaction process. Catalytic pathway for H₂ generation from methanol and H₂O catalyzed by the Pt₁-FLP dual-active sites constructed on Pt₁/PN-CeO₂.

• HAADF-STEM image and XANES spectra cannot rule out the possibility of the existence of PtO cluster. This reviewer suggests a CO-FTIR analysis. (Single Pt on CeO_x has been reported to have a very low binding energy with CO)

Response: The CO-FTIR analysis was added in the revised manuscript. As shown in Figure S8, peaks at approximately 2090 cm⁻¹ were associated with the linearly adsorbed CO on isolated ionic Pt²⁺, revealing that the Pt active sites existed as single-atom on PN-CeO₂ and coordinated with O atoms.¹⁻³ Specifically, the two apparent peaks at 2099 cm⁻¹ and 2076 cm⁻¹ could be attributed to the single-atom Pt located in spatial configuration of the type I and type II (Figure 1d), respectively, owing to the lower valence state of Pt on oxygen vacancy adjacent to the FLPs sites than it on traditional oxygen vacancy (Figure S3).

Figure S8. The DRIFTS analysis of the CO adsorption on the surface of Pt₁/PN-CeO₂.

The relevant description has been updated in the revised manuscript:

“Then, the dispersion and chemical environments of Pt on PN-CeO₂ were studied by diffuse-reflectance infrared Fourier-transform spectroscopy (DRIFTS). The peak at ~ 2090 cm⁻¹ was assigned to the linearly adsorbed CO on isolated ionic Pt²⁺ (Figure S8),¹⁻³ revealing that the Pt active sites existed as atomic dispersion on the surface of PN-CeO₂ and coordinated with O atoms. Specifically, the two apparent peaks at 2099 cm⁻¹ and 2076 cm⁻¹ could be attributed to the single-atom Pt located in the configurations of the type I and type II (Figure 1d), respectively, owing to the lower valence state of Pt on oxygen vacancy adjacent to the FLP sites than it on oxygen vacancy in the distance (Figure S3). Therefore, combining with the HAADF-STEM, XANES and DRIFTS results, the dual-active sites of single-atom Pt and FLPs were successfully constructed on the surface of Pt₁/PN-CeO₂ catalysts, where the single-atom Pt mainly occupied oxygen vacancy.” (Page 7, Line 185-196)

Reference:

1 Nie, L. *et al.* Activation of surface lattice oxygen in single-atom Pt/CeO₂ for low-temperature CO oxidation. *Science* **358**, 1419-1423 (2017).

2 Hoang, S. *et al.* Activating low-temperature diesel oxidation by single-atom Pt on TiO₂ nanowire array. *Nat. Commun.* **11**, 1062, doi:10.1038/s41467-020-14816-w (2020).

3 Lu, Y. *et al.* Unraveling the intermediate reaction complexes and critical role of support-derived oxygen atoms in CO oxidation on single-atom Pt/CeO₂. *ACS Catal.* **11**, 8701-8715 (2021).

• H/D isotope experiments and DFT calculations are well done. However, to further understand the reaction pathway of methanol-water reforming at low temperature, the authors should provide *in-situ* spectroscopic evidence, to detect/monitor the reaction intermediates.

Response: Thank you for your constructive suggestion. The mechanism of methanol decomposition over Pt/CeO₂ catalysts has been carefully investigated in the previous reports.⁴ Based on their conclusion, the CeO₂ supports exhibited no catalytic ability for methanol dissociation. While, the methoxy species adsorbed on the Pt active sites were the curtail to dehydrogenate of methanol to carbon monoxide and H₂. Herein, the Lewis acidic Ce³⁺ as a co-active site interacts with CH₃OH molecule, further promoting the CH₃OH dissociation afterwards on the single-atom Pt active sites. To experimental detect the possible assistant role of Lewis acidic Ce³⁺, the adsorption behavior of methanol on catalyst surface was explored by FTIR. As shown in Figure S21, methanol molecule only adsorbed on the oxygen vacancy of PN-CeO₂ *via* bridged methoxy species. After introduced single-atom Pt on the surface of PN-CeO₂, another characteristic peak of bridged methoxy species at 1058 cm⁻¹ was observed from the FTIR spectra. Due to the absence of bridged methoxy species on the two adjacent Ce atoms of PN-CeO₂, it could be confirmed that the methoxy species was bridged on the single-atom Pt and adjacent Ce. Therefore, the methoxy species were detected as the reaction intermediates for methanol dissociation, which was similar with previous reports.⁴

Figure S21. The FTIR spectrograms of PN-CeO₂ and Pt₁/PN-CeO₂ after adsorption of methanol.

We have added the relevant description in our revised manuscript, as following:

“To experimentally detect the roles of Lewis acidic Ce^{3+} , the adsorption behavior of methanol on the surface of $Pt_1/PN-CeO_2$ catalysts were investigated. As shown in Figure S21, methanol molecule could adsorb on the oxygen vacancy of $PN-CeO_2$ and then dissociate to methoxy species owing to the presence of the bridged methoxy species with the characteristic peak at 1033 cm^{-1} . After introducing single-atom Pt, another bridged methoxy species at 1058 cm^{-1} was appeared as revealed from the FTIR spectra. Compared with the adsorption behavior of methanol on $PN-CeO_2$, this appeared methoxy species could be attributed to the bridged configuration of methoxy on the single-atom Pt and adjacent Ce.⁴ Previous reports have proved that the CeO_2 supports exhibited no catalytic ability for methanol dissociation.⁵ Therefore, the bridged methoxy species on the single-atom Pt and adjacent Ce were detected as the critical intermediate for methanol dissociation on the $Pt_1/PN-CeO_2$ catalysts.” (Page 19, 389-400)

4 Qi, Z., Chen, L., Zhang, S., Su, J. & Somorjai, G. A. Mechanism of methanol decomposition over single-site Pt_1/CeO_2 catalyst: A DRIFTS study. *J. Am. Chem. Soc.* **143**, 60-64 (2021).

5 Chen, L. N. *et al.* Efficient hydrogen production from methanol using a single-site Pt_1/CeO_2 catalyst. *J. Am. Chem. Soc.* **141**, 17995-17999 (2019).

The dynamic structure of active sites in ceria supported Pt catalysts is very interesting research topic. Here, this reviewer suggests the authors to discuss the study of (Li, Y., Kottwitz, M., Vincent, J.L. et al. Dynamic structure of active sites in ceria-supported Pt catalysts for the water gas shift reaction. *Nat Commun* 12, 914 (2021)).

Response: Thank you for your suggestion. The preserved morphological features of the spent $Pt_1/PN-CeO_2$ catalysts could further reveal the satisfactory structure stability operated at both $165\text{ }^\circ\text{C}$ and $120\text{ }^\circ\text{C}$ (Figure S10a and S10b). Meanwhile, the surface Ce^{3+} fraction along with the $Ce^{3+}-O$ fraction of the spent $Pt_1/PN-CeO_2$ catalysts was similar with that of as-synthesized $Pt_1/PN-CeO_2$ (Figure S11), revealing that the FLPs sites on $PN-CeO_2$ was stable during the H_2 generation. After careful analysis of the HAADF-STEM images (Figure S10c), the Pt nanoclusters with small amount were observed on the surface of the spent $Pt_1/PN-CeO_2$ at $165\text{ }^\circ\text{C}$ due to the dynamical mobility of single Pt atom during the reforming of $*CO$.⁶ Also, such observed Pt clusters in the spent catalysts induced the slightly decreased H_2 generation rate of the $Pt_1/PN-CeO_2$ catalysts.

Figure S10. Morphology characterization of spent $Pt_1/PN-CeO_2$. TEM images of the used $Pt_1/PN-CeO_2$ catalysts at (a) 165 °C and (b) 120 °C, respectively. (c) HAADF-STEM image of spent $Pt_1/PN-CeO_2$ at 165 °C.

Figure S11. Surface properties of the spent $Pt_1/PN-CeO_2$ catalysts at 165 °C. XPS analysis of (a) Ce 3d and (b) O 1s peaks for the spent $Pt_1/PN-CeO_2$ catalysts.

The relevant description has been updated in the revised manuscript:

*“The preserved morphological features of the spent $Pt_1/PN-CeO_2$ catalysts could further reveal the satisfactory structure stability operated at both 165 °C and 120 °C (Figure S10a and S10b). Meanwhile, the surface Ce^{3+} fraction along with the $Ce^{3+}-O$ fraction of the used $Pt_1/PN-CeO_2$ catalysts were similar to those of as-synthesized $Pt_1/PN-CeO_2$ (Figure S11), revealing that the FLP sites were stable throughout the catalytic hydrogen generation. After careful analysis of the HAADF-STEM images (Figure S10c), the Pt nanoclusters with small amount were observed on the surface of the used $Pt_1/PN-CeO_2$ catalysts at 165 °C. Previous report has proved that the perimeter Pt active sites in the $Pt-CeO_2$ catalytic system remain dynamically mobile for the reforming of $*CO$.⁶ Thus, the decrease in H_2 generation rate of $Pt_1/PN-CeO_2$ could be attributed to the slightly increased size of Pt active sites owing to the possible mobility of the atomically dispersed Pt on the surface of catalysts. Nevertheless, combining the base-free and absence of other additives, the high H_2 generation rate and low CO selectivity as well as the satisfactory long-term stability make $Pt_1/PN-CeO_2$ featured as easy operation and high sustainability, exhibiting its great promises for practical H_2 generation from APRM at low temperatures.” (Page 12, Line 232-243)*

Nat. Commun. **12**, 914 (2021).

Reviewer #2 (Remarks to the Author):

In this present work, Sai Zhang et al. synthesized Pt single atom deposited porous CeO₂ nanorods to produce H₂ from a water and methanol mixture. The frustrated Lewis pairs of porous nanorods of CeO₂ play a vital role in methanol activation along with Pt single atoms. However, the concept of frustrated Lewis pairs is not clear and experimentally unresolved. It is tough to recommend this paper for publication at this stage.

Comments to the authors

1. The concept of frustrated Lewis pairs is not clear, and experimental evidence is not enough to support it. Since frustrated Lewis pairs is the heart of the work, it should be detailed with some experimental evidence.

Response: Thank you for your comments.

The concept of frustrated Lewis pairs (FLPs) on CeO₂ surface has been developed experimentally and theoretically in our previous reports (*Nat. Commun.* 2017, 8, 15266). Due to that the detailed constructive process of FLPs has been discussed, we lacked of some necessary description in this manuscript. To make it easier for the reader to understand, we have added the relevant illustration in our revised manuscript, as following:

As shown in Figure S1, the FLPs site is constructed by the two adjacent surface Ce³⁺ as Lewis acidic site and the neighboring surface lattice oxygen as Lewis basic site on CeO₂(110) surface.

Figure S1. (a) Optimized structure and (b) electron-density isosurface of FLPs sites on CeO₂(110) surface.

The constructive process of FLPs site has been also described in the revised manuscript. As shown in Figure 2a, for ideal CeO₂(110) surface, surface Ce and O atoms form Lewis acid-base adjuncts. The surface oxygen defect is easily formed on CeO₂ surface. When one O atom (O_{IIb}) is removed from CeO₂(110) surface, the Lewis acidic Ce_I and Lewis basic O_{IIc} atoms are expected to construct FLP sites of Ce_I-O_{IIc} after structural relaxation (Figure 2b). However, the electronic interaction between Ce_I and O_{Ia}/O_{Ic} will still hinder the activation of small molecules on the

Ce_I-O_{IIc} active sites (Figure S2e). Therefore, the weak FLPs-like activation is obtained on the traditional CeO₂ materials. When the second adjacent surface oxygen (O_{Ia}) is removed, the reduced Ce cations (Ce_I and Ce_{II}) and surface lattice oxygen (O_{IIc}) are independent Lewis acidic and basis sites (Figure S2c and S2f), respectively. Notably, two adjacent reduced surface Ce sites (Ce_I and Ce_{II}) and lattice O_{IIc} is constructed the FLPs site of (Ce_I,Ce_{II})-O_{IIc} with a shorter distance (3.99 Å).

However, the construction of surface FLPs site cannot be realized by removing surface oxygen atoms on the CeO₂(111) facet. Meanwhile, due to the low formation of oxygen vacancy on CeO₂(100) surface, the relatively unstable oxygen defect leads to the spatial configuration for formation of FLPs sites. Therefore, CeO₂(110) surface instead of CeO₂(100) and CeO₂(111) surfaces exhibits the highest possibility for FLPs construction owing to the unique FLPs configuration and formation energy of oxygen vacancies on various surfaces.

Figure S2. Schematic images of concept for design of solid frustrated Lewis pairs in CeO₂ crystal structure. (a) Optimized structure of ideal CeO₂(110). (b) Optimized structure of CeO₂(110) with one oxygen vacancy. (c) Optimized structure of CeO₂(110) with two adjacent oxygen vacancies. (d) Electron-density isosurface of ideal CeO₂(110). (e) Electron-density isosurface of CeO₂(110) with one oxygen vacancy. (f) Electron-density isosurface of CeO₂(110) with two oxygen vacancies. The electron-density isosurfaces are plotted at 0.01 e/bohr³. The color bar represents the electrostatic potential scale.

Unfortunately, it is hard to directly observe the FLP sites on the surface of CeO₂ or other reported catalysts with FLPs performance. We are constantly trying various characterization techniques to directly observe or quantify the

FLPs sites. Meanwhile, we are also actively seeking cooperation to overcome this challenge. However, so far, there is no satisfactory results. We will continue to focus on this research.

To expose the influence of FLPs sites on the catalytic performance, we have developed indirect evaluation parameter to evaluate the number of FLPs sites on various CeO₂ surface. Based on the constructive process, the adjacent oxygen vacancy is the critical to form FLPs sites on CeO₂ surface. Therefore, higher oxygen vacancy concentration inevitably results in a higher probability to form adjacent oxygen vacancy. Due to that the number of adjacent oxygen vacancy corresponds to the number of FLPs sites, the concentration of oxygen vacancy of CeO₂ is used as a parameter to indirectly describe the number of FLPs sites.

We have added this relevant description in the revised manuscript. (Supporting Information, Figure S1 and S2)

2. The d-band center of Pt atoms is one of the main factors for CO intermediate adsorption (poisoning). The d-band position should be calculated for bare Pt, Pt single atom, and Pt single atom-CeO₂ nanorods catalyst by UPS analysis.

Response: Thank you for your constructive suggestions. The d-band electron structures of various Pt sites were characterized by high-resolution valence-band XPS spectra. With the similar size of Pt nanoparticles, the d-band center of Pt nanoparticles was downward shift with the increased concentration of surface defect by comparing the Pt/NP-CeO₂, Pt/NR-CeO₂ and Pt/PN-CeO₂ catalysts (Figure S21). When the Pt active sites reduced to atomic size, the lowest d-band center (-2.65 eV) was obtained on the Pt₁/PN-CeO₂ catalysts (Figure S21), owing to the synergistic effects of the PN-CeO₂ supports with abundant defect and the atomic dispersion of Pt active sites. Thus, the significant shift of the d-band center towards the VBM resulted in a downward shift of the antibonding states of Pt active sites on Pt₁/PN-CeO₂. According to the d-band center theory, the strong bonding of adsorbates occurs if the antibonding states are shifted up relative to the Fermi level, and weak bonding occurs if the antibonding states are shifted down. Therefore, the Pt₁/PN-CeO₂ catalysts also exhibited the weakest bonding with CO molecule. However, due to the abundance of surface *OH species from FLPs sites, the rate of *CO reforming on single-atom Pt is still satisfactory compared with this transformation on Pt nanoparticles (Figure S4). In contrast, the low d-band center of single-atom Pt on PN-CeO₂ will exhibit the weak interaction with CO₂ molecule, facilitating its desorption from single-atom Pt active sites.

Figure S21. High-resolution valence-band Pt 5d XPS of Pt-x relative to the VBM, as an analogue of the density of states. Black lines indicate the positions of d-band centers.

The relevant description has been updated in the revised manuscript, as following:

“In addition, the d-band electron structures of various Pt sites were characterized by high-resolution valence-band XPS spectra. Due to the synergistic effects of the PN-CeO₂ supports with abundant defect of oxygen vacancy and the atomic dispersion of Pt active sites, the Pt₁/PN-CeO₂ catalysts exhibited the significant shift of the d-band center towards the VBM (Figure S21), resulting in a downward shift of the antibonding states. According to the d-band center theory, the strong bonding of adsorbates occurs if the antibonding states are shifted up relative to the Fermi level, and weak bonding occurs if the antibonding states are shifted down.^{7,8} Therefore, the Pt₁/PN-CeO₂ catalysts also exhibited the weakest bonding with CO₂ molecule, facilitating its desorption from single-atom Pt active sites.” (Page 19, Line 401-410)

3. DFT calculations can also be done for a d-band center shift concerning the Fermi level.

Response: Thank you for your suggestion. In order to reflect the actual catalyst, the d-band center was tested via high-resolution valence-band XPS spectra. Since we can experimentally determine the d-band center shift of various catalysts, we did not perform the DFT calculations on this issue. The detailed description has exhibited in the part of Comment 2.

4. At least CO stripping experiment should be performed by cyclic voltammetry measurement for all the synthesized catalysts.

Response: Thank you for kind suggestion. Cyclic voltammetry measurement is an effective method to detect the adsorption behavior of CO on the same catalysts *via* CO stripping experiment. Unfortunately, the electrical conductivity of the CeO₂ based catalysts is very poor, especially for the Pt₁/PN-CeO₂ catalysts with abundant surface defects. Therefore, the cyclic voltammetry measurement might not work for our catalysts.

To further confirm the adsorption of CO on Pt₁/PN-CeO₂, the CO-FTIR analysis was added in the revised manuscript. As shown in Figure S8, peaks at approximately 2090 cm⁻¹ were associated with the linearly adsorbed CO on isolated ionic Pt²⁺,¹⁻³ revealing that the Pt active sites existed as single-atom on PN-CeO₂ and coordinated with O atoms. Specifically, the two apparent peaks at 2099 cm⁻¹ and 2076 cm⁻¹ could be attributed to the single-atom Pt located in spatial configuration of the type **I** and type **II** (Figure 1d), respectively, owing to the lower valence state of Pt on oxygen vacancy adjacent to the FLPs sites than it on traditional oxygen vacancy (Figure S3).

More details have been added in the revised manuscript, as following:

*“Then, the dispersion and chemical environments of Pt on PN-CeO₂ were studied by diffuse-reflectance infrared Fourier-transform spectroscopy (DRIFTS). The peak at ~ 2090 cm⁻¹ was assigned to the linearly adsorbed CO on isolated ionic Pt²⁺ (Figure S8),¹⁻³ revealing that the Pt active sites existed as atomic dispersion on the surface of PN-CeO₂ and coordinated with O atoms. Specifically, the two apparent peaks at 2099 cm⁻¹ and 2076 cm⁻¹ could be attributed to the single-atom Pt located in the configurations of the type **I** and type **II** (Figure 1d), respectively, owing to the lower valence state of Pt on oxygen vacancy adjacent to the FLP sites than it on oxygen vacancy in the distance (Figure S3). Therefore, combining with the HAADF-STEM, XANES and DRIFTS results, the dual-active sites of single-atom Pt and FLPs were successfully constructed on the surface of Pt₁/PN-CeO₂ catalysts, where the single-atom Pt mainly occupied oxygen vacancy.” (Page 10, Line 185-196 and Figure S8.)*

Figure S8. The DRIFTS analysis of the CO adsorption on the surface of Pt₁/PN-CeO₂.

Figure 1d. The designed dual-active sites of single-atom Pt and FLPs. Type I is the dual-active sites of single-atom Pt and FLPs sites. Type II is the dual-active sites of single-atom Pt with FLPs sites. Note: The dark blue, red and yellow balls respective the Pt, O and Ce atoms, respectively.

Reference:

1 Nie, L. *et al.* Activation of surface lattice oxygen in single-atom Pt/CeO₂ for low-temperature CO oxidation. *Science* **358**, 1419-1423 (2017).

2 Hoang, S. *et al.* Activating low-temperature diesel oxidation by single-atom Pt on TiO₂ nanowire array. *Nat. Commun.* **11**, 1062, doi:10.1038/s41467-020-14816-w (2020).

3 Lu, Y. *et al.* Unraveling the intermediate reaction complexes and critical role of support-derived oxygen atoms in CO oxidation on single-atom Pt/CeO₂. *ACS Catal.* **11**, 8701-8715 (2021).

5. The surface area and porosity of CeO₂ nanorods and Pt single atom nanorods CeO₂ catalyst should be disclosed by BET analysis.

Response: The surface area and porosity of PN-CeO₂ and Pt₁/PN-CeO₂ were tested by BET analysis. The surface area of PN-CeO₂ was 109 m² g⁻¹. As shown in Figure S6b, the main pore in PN-CeO₂ was observed with pore size ~10 nm. Meanwhile, the pores with size of < 5nm was also observed. Importantly, the porous structural feature of PN-CeO₂ was strongly confirmed combining with the dark-field TEM image. After loading of single-atom Pt, the

Pt₁/PN-CeO₂ catalysts exhibited the similar nitrogen adsorption-desorption isotherm plot. The surface area of Pt₁/PN-CeO₂ increased slightly to 118 m² g⁻¹ compared with the PN-CeO₂ supports, which could be attributed to the new appeared pores with size of <2 nm. Therefore, the porous structural feature was not destroyed during loading of single-atom Pt.

The relevant description has been updated in our revised manuscript, as following:

“The specific surface area of PN-CeO₂ was 109 m² g⁻¹, as derived from the N₂ adsorption/desorption isotherm plot (Figure 6a). The porous structure with a size of 1.5~3.0 nm was revealed from TEM images (Figure 5b) as well as the Brunauer-Emmett-Teller (BET) measurements (Figure 6b and 6c). More importantly, the abundance of surface defect on PN-CeO₂ was indexed by the 30.8% surface Ce³⁺ fraction as well as the 47.1% surface Ce³⁺-O fraction, derived from its X-ray photoelectron spectroscopy (XPS) spectrum of Ce 3d and O 1s, respectively (Figure S7 and Table S1). Therefore, the FLP sites could be formed on the PN-CeO₂ supports owing to the high concentration of oxygen defect on the CeO₂(110) surface, as described in our previous reports.^{9,10}

Then, the single-atom Pt anchored on PN-CeO₂ (Pt₁/PN-CeO₂) with 0.36 wt.% loading was successfully synthesized through a photo-assisted deposition process due to the strong trapping of metal species on the defective sites of PN-CeO₂ from the DFT calculations (Figure S3). Both the specific surface area/pore structure (Figure S6) and levels of surface oxygen defects (Figure S7) of PN-CeO₂ were preserved during the photo-assisted Pt deposition process, indicating the maintained surface FLP sites.” (Page 8 Line 158-161 and Page 9 Line 171-173, highlighted by yellow)

Figure S6 | Structural characterization of PN-CeO₂ catalysts. (a) Nitrogen adsorption/desorption isotherm plot of PN-CeO₂. **(b and c)** Pore size distribution of PN-CeO₂ obtained from BET testing.

6. The wt.% of Pt atoms should be ascertained by ICP-OES measurements.

Response: Thank you for your kind suggestion. The actual loading of Pt on various catalysts have been ascertained by ICP-OES measurements. The corrected results have been updated in the Table S2.

7. The frustrated Lewis pairs in the PN-CeO₂, NR-CeO₂, and NP-CeO₂ catalyst after loading of Pt is not studied. So, it is hard to understand whether Pt influences the frustrated Lewis pairs formation or not.

Response: The surface properties of PN-CeO₂, NR-CeO₂ and NP-CeO₂ supports after loading of Pt was studied by XPS analysis, as shown in Figure S7, S11 and S12. Meanwhile, the surface fractions of Ce³⁺ and Ce³⁺-O were also summarized in Table S1. Due to the reduction of H₂, the surface defects of various CeO₂ supports exhibited a slight increase. Take Pt₁/PN-CeO₂ catalysts as example, the surface Ce³⁺ fraction increased from 30.8% to 34.7% compared with the PN-CeO₂ supports, along with the increased surface Ce⁴⁺-O fraction. Therefore, the FLPs sites on Pt₁/PN-CeO₂ supports still existed and might be increased owing to the increased surface defects concentration. Although the level of surface defects of Pt/NR-CeO₂ also increased after the Pt-loading, the surface Ce³⁺ fraction was still obviously lower than that of Pt₁/PN-CeO₂. Thus, the Pt₁/PN-CeO₂ catalysts also exhibited more amount of FLPs sites than the Pt/NR-CeO₂ catalysts. In addition, there is no FLPs sites on the surface of Pt/NP-CeO₂ due to the mismatch of crystal face.

More details could be found in Page 9, Page 13 and Page 14 (highlighted by yellow).

Figure S7. XPS analysis of (a) Ce 3d and (b) O 1s peaks for PN-CeO₂ and Pt₁/PN-CeO₂.

Figure S12. XPS analysis of (a) Ce 3d and (b) O 1s peaks for NR-CeO₂ and Pt/NR-CeO₂.

Figure S13. XPS analysis of (a) Ce 3d and (b) O 1s peaks for NP-CeO₂ and Pt/NP-CeO₂.

Table S1. Summary of surface properties of various catalysts.

Sample	Ce ³⁺ fraction (%)	Ce ³⁺ -O fraction (%)
PN-CeO ₂	30.8	47.1
Pt ₁ /PN-CeO ₂	34.7	45.4
NR-CeO ₂	19.7	34.5
Pt/NR-CeO ₂	25.6	38.4
NP-CeO ₂	15.8	23.7
Pt/NP-CeO ₂	19.2	28.7
Pt ₁ /PN-CeO ₂ -used	32.7	47.3

8. Why does a 20 % loss of catalytic performance after the stability test (Figure 3C)? Is Pt single atom leaching out from CeO₂ nanorods?

Response: After the hydrogen generation, the reaction solution was analyzed by ICP-OES. However, there was no Pt ions in the reaction solution. The surface Ce³⁺ fraction along with the Ce³⁺-O fraction of the used Pt₁/PN-CeO₂ catalysts was similar as the as-synthesized Pt₁/PN-CeO₂ (Figure S11), revealing the FLPs sites on PN-CeO₂ was stability during the H₂ generation. After careful analysis of the HAADF-STEM images (Figure S10c), the Pt nanoclusters were observed on the surface of used Pt₁/PN-CeO₂ at 165 °C. Previous report has proved that the perimeter Pt active sites in the Pt-CeO₂ catalytic system are the critical active sites for the reforming of *CO and remain dynamically mobile.⁶ Therefore, the decrease in H₂ generation rate of the Pt₁/PN-CeO₂ catalysts would be attributed to the increased size of Pt active sites owing to the possible mobility on the surface of catalysts.

We have modified the relevant description in our revised manuscript, as following:

*“The preserved morphological features of the spent Pt₁/PN-CeO₂ catalysts could further reveal the satisfactory structure stability operated at both 165 °C and 120 °C (Figure S10a and S10b). Meanwhile, the surface Ce³⁺ fraction along with the Ce³⁺-O fraction of the used Pt₁/PN-CeO₂ catalysts were similar to those of as-synthesized Pt₁/PN-CeO₂ (Figure S11), revealing that the FLP sites were stable throughout the catalytic hydrogen generation. After careful analysis of the HAADF-STEM images (Figure S10c), the Pt nanoclusters with small amount were observed on the surface of the used Pt₁/PN-CeO₂ catalysts at 165 °C. Previous report has proved that the perimeter Pt active sites in the Pt-CeO₂ catalytic system remain dynamically mobile for the reforming of *CO.⁶ Thus, the decrease in H₂ generation rate of Pt₁/PN-CeO₂ could be attributed to the slightly increased size of Pt active sites owing to the possible mobility of the atomically dispersed Pt on the surface of catalysts.”* (Page 12, Line 232-243)

Figure S10. Morphology characterization of used Pt₁/PN-CeO₂. TEM images of the used Pt₁/PN-CeO₂ catalysts at (a) 165 °C and (b) 120 °C, respectively. (c) HAADF-STEM image of used Pt₁/PN-CeO₂ at 165 °C.

Figure S11. Surface properties of the used $\text{Pt}_1/\text{PN-CeO}_2$ catalysts at 165°C . XPS analysis of (a) Ce 3d and (b) O 1s peaks for the PN-CeO₂ catalysts.

9. Besides, after the catalytic reactions, the existence of frustrated Lewis pairs sites should be studied.

Response: After the hydrogenation generation, the $\text{Pt}_1/\text{PN-CeO}_2$ catalysts were separated from the reaction solution. Then, the used $\text{Pt}_1/\text{PN-CeO}_2$ catalysts were also analyzed by XPS. As shown in Figure S11, the surface Ce^{3+} fraction was 32.7% along with the 47.3% of $\text{Ce}^{3+}\text{-O}$ fraction, similar with the values of as-synthesized $\text{Pt}_1/\text{PN-CeO}_2$ catalysts. Therefore, the absence of reduced surface oxygen defects suggested that the FLPs sites were still abundant on the used $\text{Pt}_1/\text{PN-CeO}_2$ surface.

The relevant description has been modified in our revised manuscript, as following:

“The surface Ce^{3+} fraction along with the $\text{Ce}^{3+}\text{-O}$ fraction of the used $\text{Pt}_1/\text{PN-CeO}_2$ catalysts was similar as those of as-synthesized $\text{Pt}_1/\text{PN-CeO}_2$ (Figure S11), revealing that the FLPs sites on PN-CeO₂ was stability during the H_2 generation.”

Figure S11. XPS analysis of (a) Ce 3d and (b) O 1s for the used $\text{Pt}_1/\text{PN-CeO}_2$ catalysts.

REVIEWERS' COMMENTS

Reviewer #1 (Remarks to the Author):

The authors had made a perfect revision. These new update results and discussion address the reviewer's concerns.

This review supports the publication of this study in Nature Comm.

Reviewer #2 (Remarks to the Author):

The author addressed all the comments satisfactorily and may be accepted for publication in the present form.